# Mechanism of human Lig1 regulation by PCNA in Okazaki fragment sealing

Kerry Blair[1,4], Muhammad Tehseen[2,4], Vlad-Stefan Raducanu [2], Taha Shahid[1,2], Claudia Lancey [1], Fahad Rashid[2], Ramon Crehuet [3], Samir M. Hamdan [2] ✉ & Alfredo De Biasio [1,2] ✉

During lagging strand synthesis, DNA Ligase 1 (Lig1) cooperates with the sliding clamp PCNA to seal the nicks between Okazaki fragments generated by Pol δ and Flap endonuclease 1 (FEN1). We present several cryo-EM structures combined with functional assays, showing that human Lig1 recruits PCNA to nicked DNA using two PCNA-interacting motifs (PIPs) located at its disordered N-terminus (PIP$_{N-term}$) and DNA binding domain (PIP$_{DBD}$). Once Lig1 and PCNA assemble as two-stack rings encircling DNA, PIP$_{N-term}$ is released from PCNA and only PIP$_{DBD}$ is required for ligation to facilitate the substrate handoff from FEN1. Consistently, we observed that PCNA forms a defined complex with FEN1 and nicked DNA, and it recruits Lig1 to an unoccupied monomer creating a toolbelt that drives the transfer of DNA to Lig1. Collectively, our results provide a structural model on how PCNA regulates FEN1 and Lig1 during Okazaki fragments maturation.

Genomic DNA is replicated by DNA polymerases, which only synthesize DNA in the 5′–3′ direction. As a result, one of the two antiparallel DNA strands, the lagging strand, is synthesized in short (-200 nt) Okazaki fragments needing to be enzymatically joined to produce a continuous strand[1]. In eukaryotes, Okazaki fragment synthesis begins with DNA polymerase α (Pol α) synthesizing -30 nt RNA/DNA initiator primers[2]. The homotrimeric clamp PCNA is then loaded at the primer/template junctions by the Replication Factor C clamp loader and binds the high-fidelity DNA polymerase δ (Pol δ), which processively extends the primers[3,4]. Okazaki fragments are matured by iterative cycles of Pol δ invading the previously synthesized fragments to gradually displace the initiator primers for their removal by the PCNA-bound flap endonuclease 1 (FEN1), in a process termed nick translation[1,5]. The sealing of the nicked products generated by FEN1 is performed by DNA ligase 1 (Lig1) through a 3-step nucleotidyltransferase reaction[6–8].

Lig1 comprises an unstructured N-terminal region (residues 1–262) and three folded domains (DNA binding domain, DBD; Adenylation domain, AdD; and OB-fold domain, OBD; Fig. 1a). The crystal structure of a human Lig1 fragment lacking the N-terminal region

bound to a non-ligatable adenylate-DNA substrate showed that the nicked DNA is encircled by the three Lig1 domains during catalysis[9]. The AdD (residues 536–748) and OBD (residues 749–919) constitute the catalytic core of Lig1, while the DBD (residues 263-535) provides most of the DNA affinity and aids Lig1 to encircle DNA via interactions with both the AdD and OBD[9]. The Lig1–DNA structure suggests that all three domains of Lig1 operate in concert to impose a sharp offset in the DNA duplex axis that poises the nicked DNA termini for interactions with active site residues and metal cofactors[8,9].

The interaction of Lig1 with PCNA is critical for Lig1 recruitment to sites of DNA replication and functions in the maturation of Okazaki fragments[7,10–12], while there are conflicting reports as to whether the interaction stimulates nick joining by Lig1[10,12,13]. The primary binding site of human Lig1 for PCNA encircling DNA has been mapped to a conserved PCNA-interacting motif (PIP) in the unstructured N-terminus (PIP$_{N-term}$, Fig. 1a, b)[10,11,14]. Consistently, a peptide encoding PIP$_{N-term}$ of CDC9, the yeast homolog of Lig1, has been co-crystallized with PCNA[15]. In addition, interactions between Lig1 DBD and PCNA have been reported[16,17], and a recent cryo-EM

[1]Leicester Institute of Structural & Chemical Biology and Department of Molecular & Cell Biology, University of Leicester, Lancaster Rd, Leicester LE1 7HB, UK. [2]Bioscience Program, Division of Biological and Environmental Sciences and Engineering, King Abdullah University of Science and Technology, Thuwal 23955, Saudi Arabia. [3]CSIC-Institute for Advanced Chemistry of Catalonia (IQAC) C/ Jordi Girona 18-26, 08034 Barcelona, Spain. [4]These authors contributed equally: Kerry Blair, Muhammad Tehseen. ✉e-mail: samir.hamdan@kaust.edu.sa; alfredo.debiasio@kaust.edu.sa

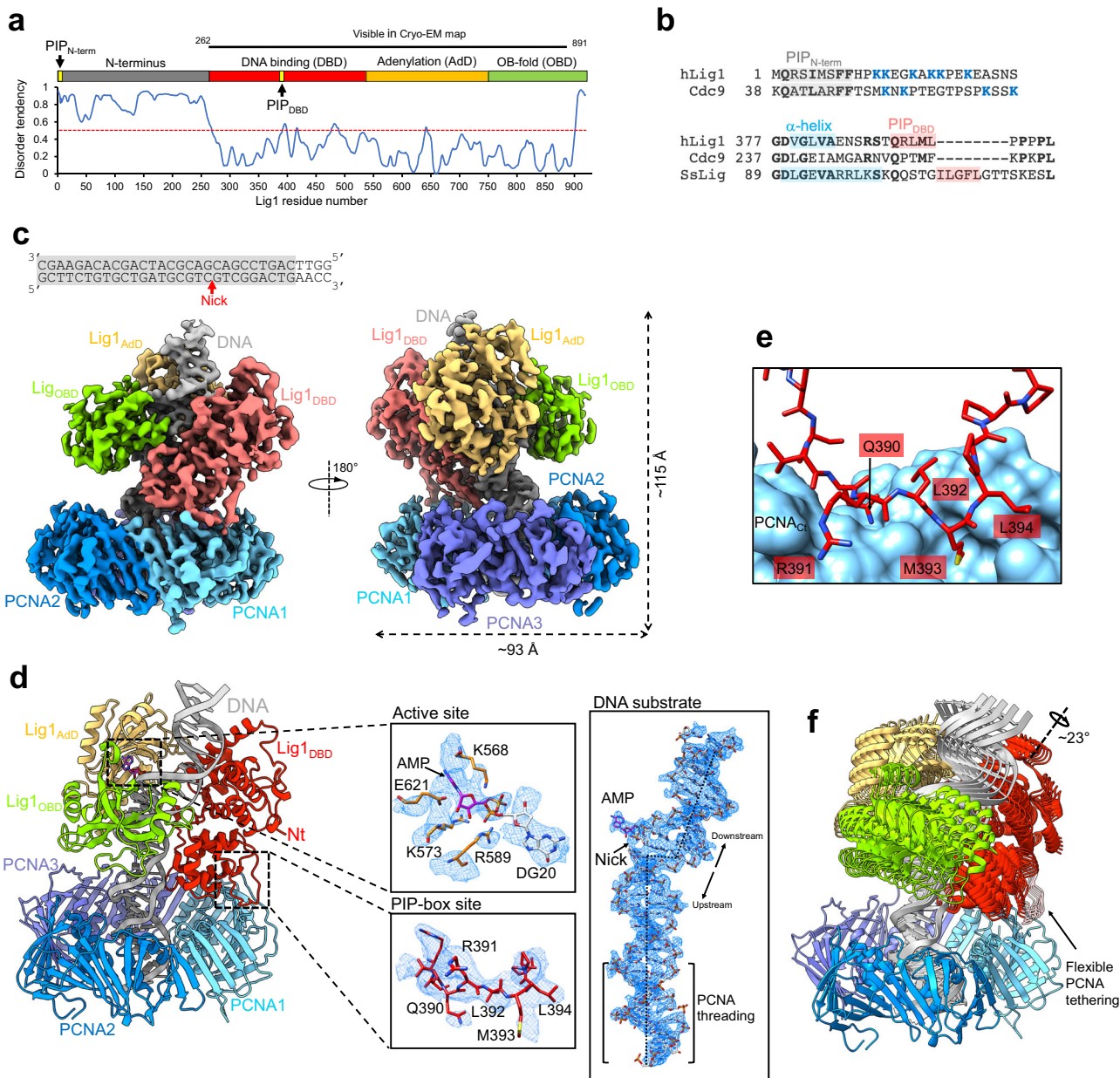

**Fig. 1 | Cryo-EM structure of the Lig1–DNA–PCNA complex. a** Disorder prediction against residue number of human Lig1, and Lig1 domain organization and position of PIP_N-term and PIP_DBD; Disorder prediction was performed with PrDOS[70]. The red dotted line corresponds to a disorder tendency of 0.5. The black line corresponds to Lig1 residues visible in the cryo-EM reconstruction. **b** Sequence alignment of ligases at the PIP_N-term and PIP_DBD regions. Residues of the canonical PIP_N-term are shaded in gray, C-terminal basic residues are colored in blue. Residues in α-helical conformation at the N-terminus of PIP_DBD in human Lig1 and ssLig[18] cryo-EM structures are shaded in blue. **c** Two views of the cryo-EM density map of the Lig1–DNA–PCNA complex colored by domains. The sequence of the nicked DNA substrate used in the study is shown. The region shaded in gray corresponds to the nucleotides modeled in the structure. The red arrow indicates the position of the

nick. **d** Side view of the model of the Lig1–DNA–PCNA complex shown in ribbon representation and colored by domains. The insets show cryo-EM map regions around different model elements depicted as sticks. The dotted line in the DNA substrate inset indicates the DNA helical axis. **e** Details of the model at the Lig1–PCNA binding site. Lig1 residues are shown as sticks, and PCNA as surface. **f** Motion represented by the first eigenvector from multi-body analysis. The first vector represents a motion involving a -23° rotation of the Lig1–DNA body around an axis encompassing the DBD longitudinally. Five positions of the Lig1–DNA body spanning the full motion are shown. This mobility indicates that the Lig1 loop interacting with PCNA is malleable to support flexible tethering of the ligase to PCNA.

structure of *S. solfataricus* (ss) ligase (which lacks a disordered N-terminal domain) bound to nicked DNA and PCNA shows the ligase tethered to PCNA via the DBD[18]. Because of the lack of structural information on the global complex of Lig1 bound to both DNA and PCNA, a defined molecular basis for how PCNA recruits Lig1 to nicked DNA is missing. Furthermore, a structural model on how PCNA coordinates the handoff of nicked DNA from FEN1 to Lig1 for nick sealing is still awaiting.

In this work, we used cryogenic electron microscopy combined with AlphaFold predictions[19], MD simulations, biophysical and functional assays to investigate how PCNA recruits full-length Lig1 to DNA nicks, and how it modulates Lig1 activity in Okazaki fragment sealing in human. Our results provide further support for a flying-cast mechanism for Lig1 recruitment[7], where the high-affinity PIP_N-term functions as an initial tether to PCNA. Our structures reveal that, once Lig1 and PCNA assemble as two stack rings encircling the nicked DNA, PIP_N-term

is released from PCNA and Lig1 stays attached to one PCNA monomer via a low-affinity PIP located in the DBD (PIP$_{DBD}$). In addition, we show that Lig1 and FEN1 form a toolbelt with PCNA in the nick sealing step of the Okazaki fragment maturation reaction, and that the PIP$_{DBD}$ tether is critical for the transfer of nicked DNA from FEN1 to Lig1 active sites to promote end joining.

## Results

### Cryo-EM structure of the Lig1−DNA−PCNA complex

We reconstituted the Lig1−DNA−PCNA complex by mixing full-length human Lig1 with a non-ligatable nicked DNA substrate bearing a dideoxy nucleotide at the upstream end of the nick, and PCNA in the presence of Mg$^{2+}$. The complex was separated by micro-size exclusion chromatography (Supplementary Fig. 1) and imaged with a Titan Krios microscope (Supplementary Figs. 2, 3). 3D classification yielded a main class that was refined to ~4.6 Å resolution, and further improved with Phenix Density Modification[20]. Lig1 and PCNA form a two-stack ring structure with approximate dimensions of 115 Å X 93 Å X 90 Å. Lig1 is suspended above the front face of the PCNA ring, with the DNA running across the ligase and through the PCNA ring hole (Fig. 1c, d). The DBD, AdD, and OBD structure and inter-domain arrangement closely resemble those described in the crystallographic study by Pascal and co-authors[9] (RMSD ~0.7 Å) (Supplementary Fig. 4). The ligase is bound to only one of the three PCNA protomers through an exposed loop located at the base of the DBD, which was not modeled in the crystal structure[9] (Fig. 1d, e). The DNA substrate is embraced by the DBD, AdD, and OBD at the nick, with the duplex segment upstream of the nick encircled by the PCNA ring. The complex architecture recalls that of ssLig bound to PCNA and DNA, and the mode of binding of the ligase DBD loop to PCNA is similar[18]. Clear density protruding from the 5′ phosphate at the nick shows that the DNA is adenylated, consistent with the previous observation that Lig1 purified from *E. coli* is fully adenylated[21,22] (Fig. 1d). The structure, therefore, represents the product of the second step of the ligation reaction, right before the joining of the nick ends.

A second dataset of the Lig1−DNA−PCNA complex reconstituted in the presence of excess ATP (Supplementary Fig. 5) yielded two 3D classes (Supplementary Fig. 6). The first class (Class1; Supplementary Fig. 6f) shows features closely resembling those of the map obtained in the absence of ATP, but no density for the OBD is observed and the map displays significant directional anisotropy. The second class (Class 2; Supplementary Fig. 6f), termed "open conformer", yielded an interpretable map at 4.2 Å resolution. Compared to the complex reconstituted without ATP, the open conformer is characterized by a different orientation of the DBD relative to PCNA, lack of density for the OBD, and poor definition of the AdD and DNA (Supplementary Figs. 6, 7). The OBD appears flexible and extending away from the AdD in 2D class averages and in a particle heterogeneity analysis using neural networks[23] (Supplementary Figs. 6, 8). Multi-body refinement[24] improved the definition of the Lig1-DNA portion of the map (Supplementary Fig. 7), revealing that the DNA downstream of the nick is loosely bound to the DBD and that the DBD rotation and the encircling of the upstream duplex DNA by PCNA force the DNA nick ends to part. The open conformer shows that nicked DNA can associate to the Lig1-PCNA complex in a partially bent conformation. The map quality, however did not allow a definitive modeling of the AdD and DNA (Supplementary Fig. 7), thus only the coordinates for the DBD and PCNA have been deposited in the PDB (Supplementary Table 1). In addition, the structure may be the product of multiple turnovers of the ligation reaction, whereby the released adenylated DNA may have rebound adenylated Lig1, and therefore its assignment to a functional state across the reaction is not possible. Because of this map's features, hereafter, we focus on the structure obtained from reconstitution in absence of ATP, which stems from a single turnover reaction.

The disordered N-terminal region of Lig1 where the primary binding site to PCNA has been previously mapped (PIP$_{N-term}$; Fig. 1a, b)[10,11,14] is invisible in the cryo-EM map (Fig. 1c), suggesting that PIP$_{N-term}$ does not participate in the interaction with PCNA once the ternary complex with DNA has formed. In the structure, the only binding site to PCNA resides in an exposed loop located approximately in the center of the DBD (residues 385-398) (Fig. 1d), and consists of an atypical PIP-box ($^{390}$QRLML$^{394}$, referred to as "PIP$_{DBD}$") which inserts between the PIP-box binding cleft of PCNA and the PCNA C-terminus (Fig. 1d, e). PIP$_{DBD}$ deviates from the conserved PIP-box (QxxΨxxϑϑ, where Ψ is an aliphatic hydrophobic residue (L, M, I, V), ϑ is aromatic, and x can be any amino acid)[25] but shows some similarities in the mode of binding to PCNA[26]: the side chain of Gln (position +1) binds to the so-called Q-pocket and that of Met (position +4) fits into the canonical hydrophobic pocket (Fig. 1d). The lack of aromatic residues at positions +7 and +8 in PIP$_{DBD}$ points to a rather weak interaction, and the resulting small buried interface between Lig1 and PCNA (~790 Å$^2$) is expected to confer local dynamics to Lig1. In agreement, the resolution of the Lig1−DNA map portion was improved by multi-body refinement[24] (Supplementary Fig. 2), indicative of its mobility relative to PCNA. From the positional variance derived from principal component analysis of the multi-body refinement[24], motion along the first eigenvector accounts for ~20% of the flexibility (Supplementary Fig. 2), and is characterized by a monomodal distribution of amplitudes, suggestive of continuous motion between the Lig1−DNA and PCNA bodies, involving a ~23° rotation of Lig1 around an axis encompassing the DBD longitudinally (Fig. 1f). Such mobility indicates that the loop containing PIP$_{DBD}$ is malleable and able to tether the ligase to PCNA in various orientations.

In the archaeal Lig−DNA−PCNA cryo-EM structure[18], a second non-PIP region in the DBD core contacts PCNA extensively; the corresponding region in human Lig1 is positioned further away from PCNA (with a buried surface of only ~195 Å$^2$), and the side chains at the interface are poorly ordered, suggesting that the interaction may not be conserved. Similarly, the interaction between the AdD and PCNA reported in the archaeal complex[18] is not observed in human, where the distance between the AdD and PCNA is >10 Å.

The DNA portion of the map is well defined and could be modeled unambiguously (Fig. 1d). Superposition of the cryo-EM model to the X-ray structure of the Lig1−DNA complex[9] demonstrates that the DNA features are equivalent (Supplementary Fig. 4). These features include a large (>5 Å) offset of the DNA axis at the nick, A-form conformation for the duplex immediately upstream of the nick, and B-form conformation for the downstream DNA. The map region at the AMP binding pocket shows that the DNA is adenylated and the active site residues are partly ordered (Fig. 1d). The map resolution, on the other hand, prevented a reliable discrimination of metal cofactors bound to the active site. The duplex DNA upstream of the nick traverses the center of the PCNA ring hole in B-conformation, with its axis almost perpendicular (~4° tilt) to the ring plane and without direct contacts with residues of the ring inner rim (Fig. 1d). The network of interactions between the three Lig1 domains and DNA is equivalent to that described by Pascal and co-authors[9]. The DBD provides most of the contacts, particularly with the minor groove, where a long loop binds the strand downstream of the nick, and one turn between two α-helices engages the strand upstream of the nick, primarily through main-chain interactions. The AdD appears to stabilize the DNA structure via interactions mediated by the AMP cofactor, with additional sparse contacts with the DNA backbone.

### Role of the N-terminal region of Lig1 in PCNA recruitment

In light of the cryo-EM structure (Fig. 1) and previous observations[10,11,14], it appears that PIP$_{N-term}$ may function as an initial tether to PCNA, which becomes dispensable once the ternary complex with DNA has formed. To gain insight into the arrangement of the Lig1−PCNA complex in the absence of DNA and the possible role of the N-terminal

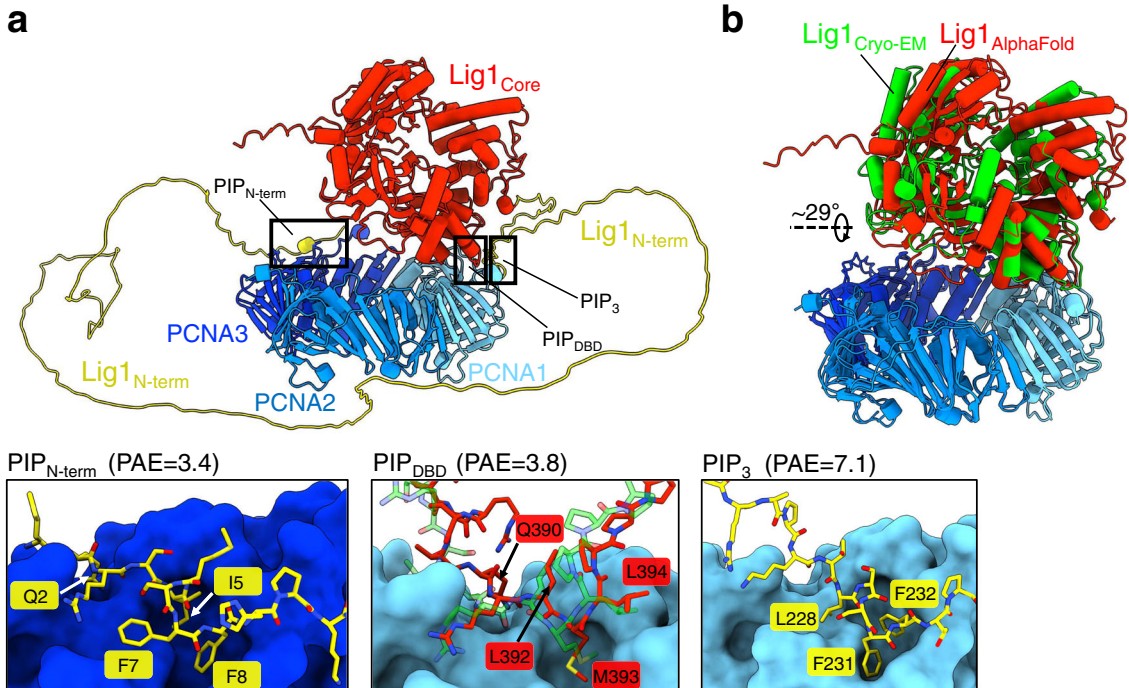

**Fig. 2 | AlphaFold prediction of the Lig1–PCNA complex. a** First ranked model showing the Lig1 N-terminal region (residues 1-263) and Lig1 Core (residues 264-919) as yellow and red ribbons, respectively, and PCNA trimer as ribbons in different shades of blue. The insets show close-ups of the three predicted binding sites and associated aligned error (PAE), with Lig1 residues shown as sticks and PCNA as surface. The middle panel shows an overlay of the predicted PIP_DBD interface (red sticks) and that from the cryo-EM structure (green sticks). The high PAE, lack of conservation and absence of experimental evidence of binding for PIP3 suggest that PIP3 is a spurious prediction. **b** Overlay of the AlphaFold model and cryo-EM structure on PCNA. The positions of the Lig1 Core are related by a 29° rotation around the indicated axis. The N-terminal region of the AlphaFold model and DNA in the cryo-EM model are omitted for clarity.

region, we carried out a prediction with AlphaFold[19], and compared it with the cryo-EM structure (Fig. 2). Strikingly, the first ranked model predicts an architecture very similar to the cryo-EM model, with Lig1 positioned above the PCNA ring (Fig. 2a) and Lig1 core only slightly (~29°) rotated (Fig. 2b). The Lig1 core is attached to one PCNA monomer via the PIP_DBD in a mode virtually identical to the cryo-EM structure (Fig. 2a, middle inset). The Lig1 N-terminal region is predicted as an extended chain winding around the core, with its conserved PIP-box tethered to a second PCNA monomer (Fig. 2a, left inset), in a mode analogous to that observed in the X-ray structure of a PIP_N-term peptide of CDC9 bound to PCNA[15]. A third atypical PIP box (PIP3), adjacent to the DBD C-terminus, co-exists with PIP_DBD at the same PCNA pocket (Fig. 2a, right inset). The high Predicted Aligning Error (PAE), the lack of conservation and the absence of experimental evidence of binding activity for PIP3[10,11] suggest that PIP3 is a spurious prediction.

It is likely that, before PIP_DBD is engaged, PIP_N-term tethers Lig1 to PCNA in an ensemble of orientations due to the conformational flexibility of the N-terminal domain. In the AlphaFold prediction, which pertains to a state post PIP_DBD engagement, the flexible N-terminus extending from the Lig1 core needs to fold back to stay attached to the second PIP pocket on PCNA: it is possible that the entropic penalty associated with this chain reversion results in the release of PIP_N-term once the Lig1 core has assembled around the DNA (Fig. 1).

To further probe the role of PIP_N-term and PIP_DBD in Lig1 recruitment to PCNA, we compared the PCNA-binding activity of wild-type Lig1 (WT_Lig1) and Lig1 variants in which the entire N-terminus was deleted (ΔN_Lig1, residues 233-919) or PIP_DBD residues were mutated to alanines (LML_Lig1 and QRLML_Lig1), in the absence and presence of nicked DNA. As expected from previous findings[10,11], in the absence of DNA the PIP_DBD mutant retains high-affinity binding to PCNA ($K_d$ ~ 45 nM), suggesting that the *off-DNA* interaction with PCNA is primarily mediated by PIP_N-term

(Fig. 3a–c). The binding affinity of the PIP_N-term is likely increased by the stretch of basic residues adjacent to the PIP (Fig. 1b), similar to other high-affinity PIPs[25]. To test the effects of the N-terminal deletion and DBD mutations in the presence of DNA, we monitored the retention of PCNA on a nicked DNA pre-bound to WT_Lig1 or Lig1 variants by measuring the Förster Resonance Energy Transfer (FRET) between Cy3-labeled PCNA and Alexa Fluor 647- labeled DNA (Fig. 3d). The nicked DNA contained a dideoxy nucleotide at the 3' of the nick to prevent ligation and substrate release. In the absence of Lig1, no increase in Alexa Fluor 647 emission was observed above the baseline level corresponding to the direct sum of the fluorophores' emission (Fig. 3e), consistent with no retention of PCNA-Cy3. Upon addition of WT_Lig1 or Lig1 variants, different increases in Alexa Fluor 647 emission were observed, anti-correlated with a decrease in Cy3 emission, demonstrating that the Alexa Fluor 647 emission increase stems from a FRET mechanism (Fig. 3e). Nevertheless, we could not calculate an exact FRET efficiency due to the PCNA labeling stoichiometry. Thus, we deconvoluted each emission spectrum into its Cy3 and Alexa Fluor 647 spectral contributions (Fig. 3f). The deconvoluted amplitude of the contribution of Alexa Fluor 647 emission was recorded for each experimental condition. The efficiency of PCNA retention decreases for all Lig1 variants, with the N-terminal deletion showing the most dramatic effect, followed by QRLML_Lig1 and LML_Lig1; when combined, the N-terminal deletion and QRLML mutation reduce retention nearly to the levels measured in the absence of Lig1 (Fig. 3f). Thus, both Lig1 N-terminal and DBD interactions contribute to the recruitment of PCNA onto nicked DNA in this assay. A similar conclusion is drawn by cryo-EM imaging of a mixture containing ΔN_Lig1, nicked DNA, and PCNA previously separated by micro-SEC (Supplementary Figs. 1, 9). The SEC peak elutes later compared to the

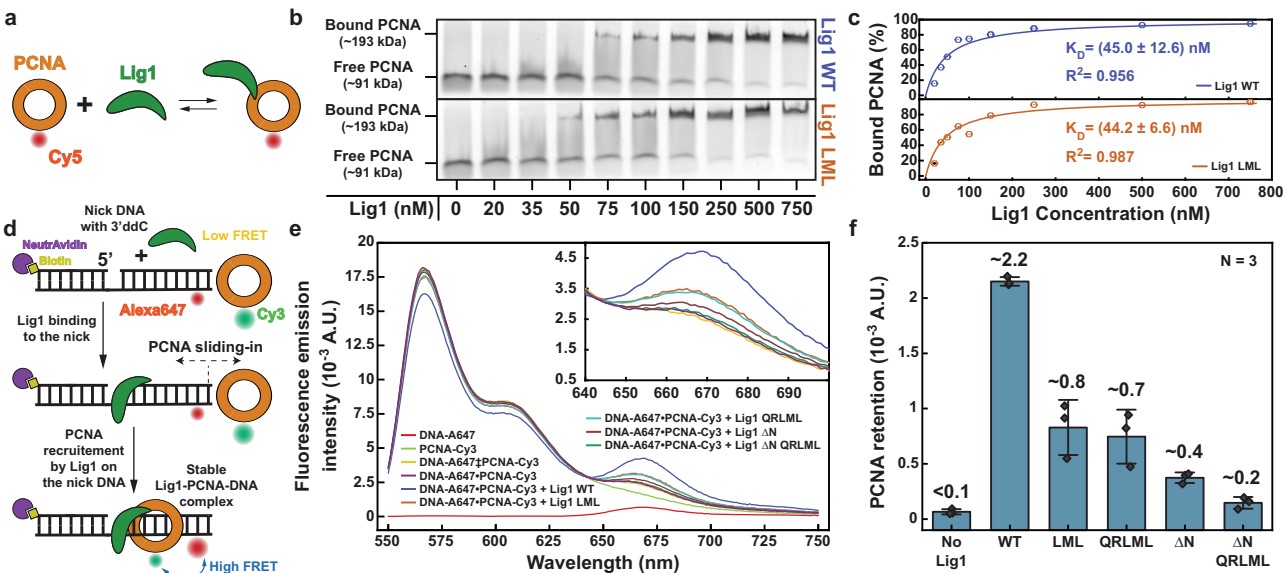

**Fig. 3 | Biochemical assays of Lig1 binding to PCNA in absence and presence of DNA. a** Schematics of the experiment shown in panel (**b**). **b** Protein–protein EMSA was used to monitor the binding of WT_Lig1 and LML_Lig1 to Cy5-labeled PCNA (1 nM) in solution in the absence of DNA. **c** Quantification of the EMSA data presented in panel (**b**). The experimental data points were fitted to quadratic dependencies as described in "Methods". **d** Schematics of the experiment shown in panel (**e**). **e** Emission spectra of DNA-Alexa Fluor 647 and PCNA-Cy3 in the absence and presence of WT_Lig1 or Lig1 variants. DNA-Alexa Fluor 647 (500 nM) was blocked with NeutrAvidin (1 μM) at the end of the downstream arm to prevent PCNA sliding-in from that end and to allow for PCNA sliding-in only through the 3′ duplex arm. The nicked DNA contained a 3′ ddC to prevent ligation. PCNA-Cy3 (500 nM; 3:1 = Cy3:PCNA) was then added. The direct sum (‡) of PCNA-Cy3 and DNA-Alexa Fluor 647 fluorescence emission intensity (upon excitation at 480 nm) is used as a baseline. The inset shows a closer view of the Alexa Fluor 647 emission band. **f** Quantification of the data presented in panel **e**. The emission spectra of the baseline (‡), PCNA-Cy3-DNA-Alexa Fluor 647 in the absence of Lig1 and in the presence of WT_Lig1 or Lig1 variants were fitted with a linear combination of Cy3 and Alexa647 emission spectra. The deconvoluted amplitude of the Alexa Fluor 647 emission contribution was recorded for each of the four spectra. The value corresponding to the baseline condition (‡) was subtracted from the other six conditions. The bar chart shows the quantified increase in Alexa Fluor 647 emission, above the baseline, for the six conditions. The bar chart illustrates the mean (as bar height) and one standard deviation (as error bar) of $N = 3$ independent measurements. Source data are provided as a Source Data file.

---

mixture with wild-type Lig1 (Supplementary Fig. 1), consistent with a reduced hydrodynamic radius and/or with a partial dissociation of the complex. Indeed, 2D class averages from the eluted peak show a high proportion of dissociated PCNA, and a low proportion of formed ternary complex (Supplementary Fig. 9).

Collectively, our structural and binding data argue that the interaction mediated by the Lig1 N-terminus facilitates the initial recruitment of PCNA from solution, and that the interaction with the DBD stabilizes the functional complex on nicked DNA.

## PCNA modulates Lig1 and FEN1 function in Okazaki fragments sealing

We confirmed the previous finding[10,13] that the presence of PCNA does not affect the activity of Lig1 in ligating normal nicked substrates (Supplementary Fig. 10a), possibly because the affinity of the ligase for a free nick in a short substrate is sufficiently strong to bypass the need of the PCNA tether. In this context, the interaction of Lig1 with PCNA tethers Lig1 to the DNA substrate but does not significantly improve the ability of Lig1 to locate the nick within the short DNA molecule. This is in contrast with observations in the archaeal system[18], where PCNA stimulated ssLig activity in vitro. The differences in archaeal and eukaryotic ligase topology as well in the ligases' interaction with PCNA may in part explain this discrepancy. Because PCNA participates with both Lig1 and FEN1 in the maturation of Okazaki fragments[1], we formed the hypothesis that PCNA may modulate the ligation activity of Lig1 when FEN1 is present in the reaction.

During Okazaki fragment processing, the strand displacement activity of Pol δ creates 5′ flaps in previously synthesized fragments that are cleaved by FEN1, to generate ligatable nicks[27]. FEN1 actively bends flap-DNA by ~100° and threads the 5′ flap through the hole created by the so-called cap-helical gateway, guiding it into the cleavage site[28-31]. FEN1 association to PCNA, which occurs via a single PIP-box at the FEN1 C-terminus[32], increases FEN1 affinity for the flap substrate without affecting the catalytic step[33]. The nick product is released by FEN1 in two steps, where it is briefly retained in a bent conformation followed by a lengthy binding in a more extended conformation[34,35]. We thus tested whether Lig1 may need to actively displace the nicked product from PCNA-bound FEN1 to complete nick sealing, and whether the Lig1–PCNA interaction may modulate the substrate handover when both enzymes are simultaneously bound to one PCNA ring. For this purpose, we monitored the multiple-turnover ligation kinetics of wild-type Lig1 or Lig1 variants in the presence of FEN1 that was preassembled with nicked DNA and PCNA (Fig. 4a and Supplementary Fig. 10b). A FEN1 inactive mutant (D181A) was used to suppress FEN1 exonuclease activity[34]. The initial ligation rates of the LML_Lig1 and QRLML_Lig1 mutants were found to be ~3-fold slower than the wild-type (Fig. 4b, c). These results corroborate the binding data (Fig. 3) to show that the Lig1–PCNA interaction via the DBD stabilizes the Lig1–DNA–PCNA ternary complex and demonstrate that this stabilization becomes critical when FEN1 is present and Lig1 needs to compete for engaging the nicked DNA. Interestingly, the ligation rate of ΔN_Lig1 is similar to that of wild-type Lig1 (Fig. 4d), suggesting that the interaction of $PIP_{N-term}$ with PCNA becomes dispensable when PCNA is stably loaded by FEN1 on the nicked DNA and it does not need to be recruited from solution.

## PCNA forms a toolbelt with Lig1 and FEN1

We next focused on providing structural information as to how PCNA may coordinate the activity of FEN1 and Lig1 during maturation of Okazaki fragments. We started by focusing on understanding the interaction of FEN1 with PCNA in the presence of nicked DNA. To this

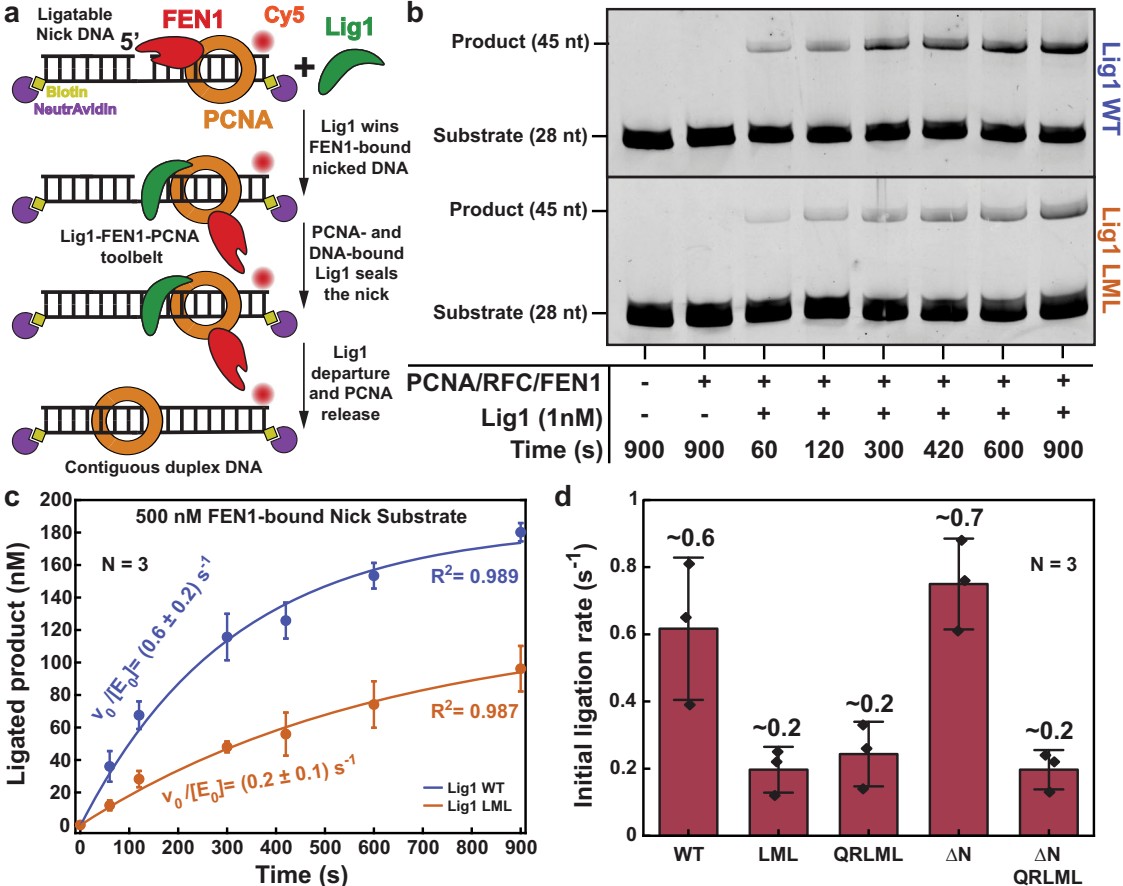

**Fig. 4 | PCNA interaction-dependent modulation of Lig1 ligation rates in the presence of FEN1. a** Schematics of the experiment shown in panel (**b**). **b** Multiple-turnover ligation by Lig1 in the presence of trapped PCNA and competitor FEN1. A double-blocked (NeutrAvidin bound to both DNA termini; 1 μM NeutrAvidin) DNA nick substrate (500 nM) was used for this experiment. PCNA (500 nM) was loaded on this substrate by RFC (500 nM). FEN1 D181A (500 nM) was added and incubated with the PCNA-loaded substrate. The reaction was initiated by adding WT_Lig1 or LML_Lig1 (1 nM) and incubated for the indicated amount of time at 37 °C. **c** Quantification of the data from panel (**b**) as described in the Methods section. The single-exponential burst parameters are irrelevant kinetically and only their derivative near $t = 0$ is employed to estimate the normalized initial rate. For WT_Lig1 the burst parameters were A = (183.9 ± 25.2) nM and $\tau_{obs}$ = (311.4 ± 103.5) s. For LML_Lig1 the burst parameters were A = (123.3 ± 38.0) nM and $\tau_{obs}$ = (625.8 ± 324.6) s. The experimental data points represent the mean and one standard deviation of $N = 3$ independent reactions. **d** Bar chart showing the initial ligation rates of WT_Lig1 and Lig1 variants quantified from the derivative of the bursts presented in Fig. 4c and Supplementary Fig. 11c near $t = 0$. The bar chart illustrates the mean (as bar height) and one standard deviation (as error bar) of $N = 3$ independent reactions. Source data are provided as a Source Data file.

end, we imaged by cryo-EM a mixture of FEN1, PCNA and the nicked DNA substrate used for reconstitution of the Lig1 complex (Supplementary Fig. 11). We obtained an intermediate resolution (~7.8 Å) reconstruction of the complex, showing FEN1 attached to PCNA through the C-terminal PIP-box and occupying an upright position on the front face of the PCNA ring (Fig. 5a, b). The path of the DNA axis bends ~100° at the location of the nick, in agreement with the X-ray structure of FEN1 bound to a 5′-flap DNA substrate post flap cleavage[28,30], while the upstream duplex DNA extends in B-form centrally through the PCNA pore (Fig. 5b). The lack of density corresponding to the two helices of the cap/helical gateway suggests that the helices are flexible (Fig. 5b). The resolution of the FEN1−DNA portion of the map is lower than the average resolution (Supplementary Fig. 11), supporting flexibility of the FEN1−DNA subcomplex, as predicted by previous MD simulations[36].

In the FEN1−DNA−PCNA complex, the nicked DNA is exposed, suggesting that Lig1 may capture it by attaching to one of the two vacant PIP binding sites on PCNA. To test this possibility, we determined the cryo-EM structure of Lig1 reconstituted with PCNA, FEN1, and a nicked DNA substrate in the presence of ATP and Mg²⁺ (Supplementary Figs. 12, 13). Data were acquired with 0° and 30° tilting of the specimen stage to improve the angular distribution of particles,

yielding a map at a global resolution of ~4.4 Å (Supplementary Fig. 13). All complex components in the map were readily assigned and could be modeled (Fig. 5c, d). Lig1 associates with PCNA and DNA in a mode analogous to that observed in the absence of FEN1, but the Lig1−DNA body is slightly rotated (~12°) to accommodate FEN1. FEN1 binds PCNA in an upright configuration, through the interaction between the FEN1 C-terminal PIP-box and the most exposed of the two PIP-box sites not occupied by the ligase, and is positioned in close proximity to the OBD. Compared to the DNA-bound form (Fig. 5a, b), FEN1 is rotated ~17° around its C-terminal hinge. Only the central part of FEN1 core appears rigid, while the regions corresponding to the cap/helical gateway and hydrophobic wedge are invisible, suggesting that they are flexible. The most ordered region of the FEN1 map portion corresponds to a wide groove created by helices αH1, αH2 and strands β1, β6 and β7 of the twisted β-sheet core[28], which is approached by a loop connecting two strands of the OBD β-barrel (Fig. 5e). The partial disorder of FEN1, the lack of a constitutive FEN1−OBD interface (~500 Å² of buried surface), and the mobility of the OBD relative to FEN1 shown by multi-body refinement (Supplementary Fig. 13), point to the absence of a direct interaction between Lig1 and FEN1 in the complex. In addition, our MD simulation probing diverging OBD conformations (Supplementary Movie 1), mimicking a step of the end joining reaction prior to DNA

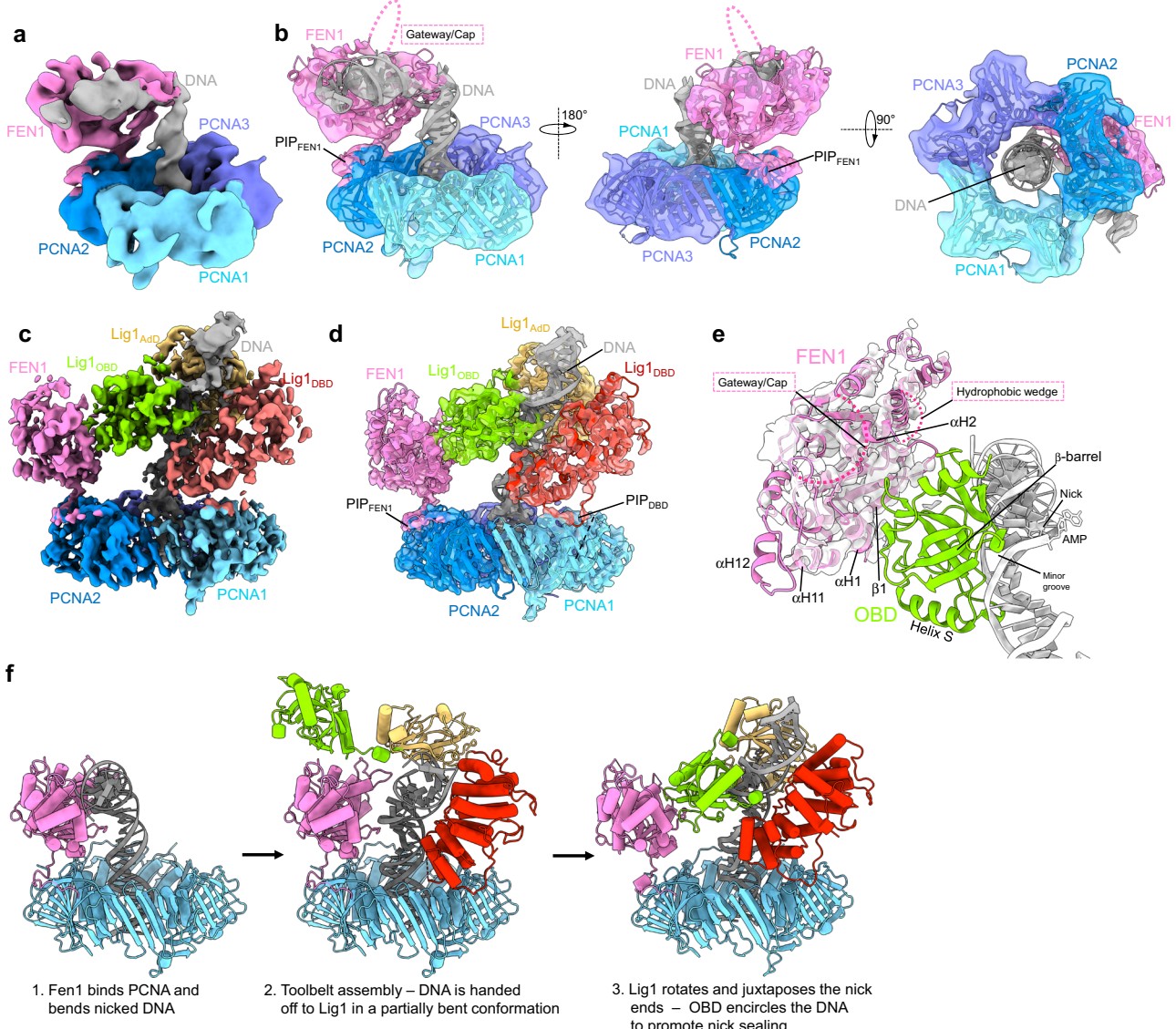

**Fig. 5 | Cryo-EM structures of FEN1–DNA–PCNA and Lig1–DNA–PCNA–FEN1 complexes. a** Cryo-EM density map of the FEN1–DNA–PCNA complex colored by domains. **b** Three views of the cryo-EM density map of the FEN1–DNA–PCNA complex with fitted model. **c** Two views of the cryo-EM density map of the Lig1–DNA–PCNA–FEN1 complex colored by domains. **d** Model of the complex in ribbon representation fitted into the cryo-EM map, and colored by component. The structure represents a step of the Okazaki fragment maturation reaction following handover of the nicked DNA substrate from FEN1 to Lig1 to complete nick sealing. **e** Details of the model at the FEN1–OBD interface. The cryo-EM map around FEN1 is shown, indicating that only the FEN1 central core is well ordered. **f** Possible steps for the mechanism of handoff of nicked DNA from FEN1 to Lig1. The model in the central panel shows Lig1 in the "open conformation", where the OBD is flexible and extending away from the AdD (Supplementary Fig. 7).

encirclement by the OBD, suggests that FEN1 binding to PCNA does not restrict the conformational space sampled by the OBD (Supplementary Fig. 14).

In summary, the toolbelt structure represents a step of the Okazaki fragment processing reaction following the handover of the nicked DNA substrate from FEN1 to Lig1 to complete nick sealing, and agrees with the activity assays showing that binding of Lig1 to PCNA through PIP$_{DBD}$ facilitates the seizing of nicked DNA by the ligase. While the details of the handover mechanism are unknown, it is possible that Lig1 captures the nicked DNA in a partially bent conformation (Fig. 5f), analogous to that observed in the open conformer of the Lig1–DNA–PCNA complex reported here (Supplementary Fig. 7), where the flexible OBD would make the AdD-DBD subcomplex accessible for DNA transfer. A rotation of the DBD relative to PCNA would then straighten the DNA and poise it to be encircled by the OBD in the last step of nick sealing (Fig. 5f).

## Discussion

In this work, we dissected the molecular determinants of human Lig1 recruitment to PCNA using structural, computational, and biophysical approaches. Taken together, our results provide a mechanism for Lig1 recruitment, and reconcile previous biochemical and cellular observations[10,11,14]. The canonical, high-affinity PIP motif located at the extreme of the disordered Lig1 N-terminal region (PIP$_{N-term}$) functions as a tether to PCNA when the ligase is detached from DNA (Fig. 6a), and may facilitate the efficient scan of nicks due to the extremely fast 1D diffusion of PCNA along duplex DNA (~8 base pairs per microsecond[37]). Once a nick is encountered, PIP$_{N-term}$ is released and Lig1 binds PCNA through a low-affinity PIP located in the DBD (PIP$_{DBD}$), which dynamically holds the ligase onto the nicked DNA and on top of the PCNA ring to promote nick sealing (Fig. 6b). The reason why PIP$_{N-term}$ is released upon complex formation with nicked DNA is unclear, but it may be caused by the entropic penalty resulting from the binding of the Lig1

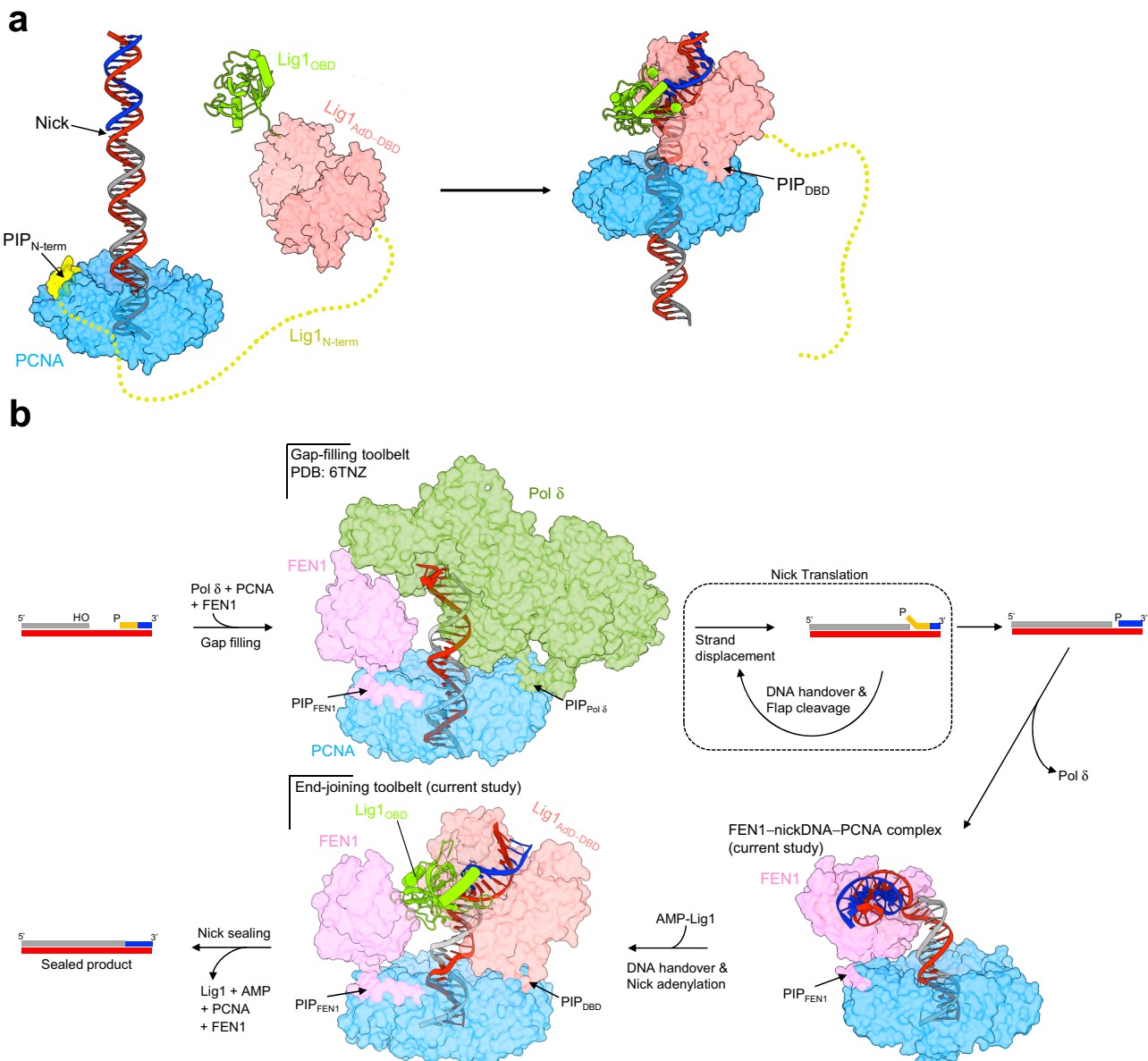

**Fig. 6 | Proposed model of function of human Lig1 in nick sealing. a** Mechanism of PCNA-directed Lig1 recruitment to nicked DNA. Away from a nick site, the high-affinity PIP in the disordered Lig1 N-terminus tethers Lig1 to PCNA. Once a nick is encountered, the N-terminal PIP is released and Lig1 binds PCNA through a low-affinity PIP located in the DBD, which dynamically holds the ligase onto the nicked DNA and on top of the PCNA ring to promote nick sealing. **b** Proposed structural basis for the PCNA-directed function of Pol δ, FEN1 and Lig1 in Okazaki fragment maturation in human. In the first step of the reaction, Pol δ and FEN1 assemble around the primer-template junction of an Okazaki fragment, forming a toolbelt with PCNA, and Pol δ fills the gap between adjacent fragments[42]. Once the 5′-end of a previously synthesized fragment is reached, the strand-displacement activity of Pol δ replicates through the initiator primer (colored in orange), generating a 5′ flap. The substrate is handed off to FEN1 for flap cleavage and the process repeats iteratively until full removal of the primer (nick translation), when Pol δ dissociates from PCNA. The resulting nicked DNA remains associated with FEN1 until the incoming Lig1 binds PCNA via the PIP-box located in the DBD, forming a second toolbelt with FEN1. Lig1 then captures and fully encircles the substrate to complete nick sealing.

catalytic core to PCNA, which would require a folding of the flexible N-terminus around the core to allow PIP$_{N-term}$ to stay attached to a second PIP binding site.

Our structures of Lig1 bound to nicked DNA and PCNA highlight the conformational flexibility of the 3-domain ligase, and the plasticity of the PCNA-ligase interface. These properties appear integral for Lig1 function in critical phases of Okazaki fragment sealing. Studies on human and archaeal DNA ligases[9,17,38] showed that the ligase OBD must rotate >90° after catalyzing Step 1 of DNA end-joining (ligase adeny-lation) to accomplish the DNA-dependent steps of the reaction (Steps 2 and 3). Our data on the open conformer of the Lig1−DNA−PCNA complex, showing a flexible OBD, agree with structural findings

reporting OBD flexibility in archaeal ligases bound to DNA and PCNA[18,39]. In addition, the open conformer shows that nicked DNA can associate to the Lig1−PCNA complex in a partially bent conformation, with the nick ends set apart. While the role of the open conformer remains unclear, it may facilitate the handoff of the DNA from FEN1 in Okazaki fragment maturation (further discussed below), or to a repair enzyme in case the nick ends are damaged and not ligatable.

Because of PCNA's trimeric structure, it is possible that multiple enzymes may simultaneously bind to PCNA, each occupying a different monomer; in Okazaki fragment maturation, this so-called "toolbelt model" has been observed in the archaeal system[40]. The alternative model, which assumes dynamic binding to and dissociation from

PCNA is named the "sequential model". Biochemical experiments in the yeast system[41] using mutated PCNA heterotrimers, showed that nick translation and sealing do not strictly require simultaneous binding of Pol δ, FEN1 or Lig1 to PCNA, but the methodology could not ascertain whether a sequential switching of PCNA partners indeed occurs. Conversely, a biochemical report characterizing Okazaki fragment processing at the millisecond time scale supported the formation of a yeast quaternary Pol δ−DNA−PCNA−FEN1 complex performing processive nick-translation synthesis[5]. Our own structural work[42] showed that a human Pol δ−DNA−PCNA−FEN1 complex actually forms, adding further weight to the toolbelt model in nick translation.

To complete nick sealing, Lig1 must capture the nicked product generated by FEN1. Here, we reported an intermediate resolution cryo-EM structure of FEN1 bound to nicked DNA and PCNA, showing that FEN1 binds one of the three PCNA protomers and grips the DNA sharply bent at the nick in an exposed position above the front face of the PCNA ring, accessible to an incoming binding partner. Indeed, we show that Lig1 binding to an unoccupied PCNA protomer of the FEN1−DNA−PCNA complex via Lig1 PIP_DBD actively drives the transfer of DNA to the ligase active site. Presumably, the flexibility of the OBD domain of the ligase facilitates the intermediate step of substrate handoff, when the DBD-AdD subcomplex needs to be open and accessible for DNA transfer. It is likely that substrate handover occurs via ligase conformation sampling aided by the flexible tethering of the ligase to PCNA and the juxtaposition of the DNA by FEN1 in the toolbelt. The ability of Lig1 to accommodate small differences of bending of nicked DNA may further facilitate the handoff from FEN1. Once the DNA is embraced by the DBD-AdD, the OBD can stably encircle the DNA to promote nick sealing.

We have previously solved the structure of the Pol δ−DNA−PCN −FEN1 toolbelt captured during the gap-filling step and prior to the strand displacement step that generates the flap substrate[42] (Fig. 6b). In this toolbelt, FEN1 binds the PCNA protomer that is opposite to Pol δ and poised for the handoff of the flap substrate produced by Pol δ. Interestingly, FEN1 and Lig1 are oriented similarly in the Lig1−DNA −PCNA−FEN1 toolbelt (Fig. 6b). In both toolbelts the binding site on the third PCNA protomer is significantly less exposed to support the simultaneous binding of Pol δ and Lig1 with FEN1. Our results, backed by recent biochemical experiments[43], suggest that maturation of Okazaki fragments in human is likely to function via two toolbelts that are centralized around FEN1: the Pol δ−PCNA−FEN1 toolbelt that mediates the handoff of the flap substrate from Pol δ to FEN1 and, upon FEN1 cleavage of the flap substrate, the Lig1−PCNA−FEN1 toolbelt that transfers the nicked DNA from FEN1 to Lig1 for its ligation (Fig. 6b).

In addition, the orientation of the duplex DNA in the central channel of PCNA appears similar in both toolbelts and in the FEN1−DNA−PCNA complex. Collectively, these results suggest that PCNA facilitates the transfer of the products among its binding partner proteins passively via protein−protein interaction rather than by actively orienting the DNA. In this mechanism, maintaining the DNA orientation by PCNA might also create more steric hindrance among the partner proteins and force the formation of multiple toolbelts depending on the functionality mediated by PCNA. Our structural findings pave the way for further studies to characterize the communication between Pol δ, FEN1, Lig1 and PCNA during the maturation of Okazaki fragments and how PCNA acts as a toolbox during DNA replication, repair and recombination.

## Methods
### DNA substrates
DNA oligos for the ligation and fluorescence experiments were synthesized and HPLC purified by IDT. Sequences of the oligonucleotides are listed in Supplementary Table 2. The substrates for Lig1 multiple-turnover kinetics assays and steady-state fluorescence retention experiments were generated by annealing oligos in a 1:1 molar ratio in

TE-100 buffer [50 mM Tris·HCl (pH 8.0), 1 mM EDTA and 100 mM NaCl]. The mixture was heated at 95 °C for 5 min followed by slow cooling down to room temperature. The annealed product was PAGE purified to >90% purity using 10% non-denaturing polyacrylamide gel electrophoresis (Invitrogen). DNA oligos for cryo-EM analysis were synthesized and HPLC-purified by Sigma Aldrich. The nicked DNA substrate consisted of a 32 nt template strand oligo (Oligo32), a 19 nt oligo with a dideoxy cytosine at the 3′ end (Oligo_19ddC), and a 13 nt oligo with a phosphate group at the 5′ end (Oligo_13P). The sequences of these three oligos are: Oligo32; 5′-GGTTCAGTCCGACGACGCAT-CAGCACAGAAGC. Oligo_19ddC; 5′-GCTTCTGTGCTGATGCGT[23ddC]. Oligo_13P; 5′-[P]GTCGGACTGAACC. The nicked DNA substrate was annealed by mixing the oligos in an equimolar ratio in the presence of 20 mM Tris·HCl (pH 7.5) and 25 mM NaCl. The oligos were then annealed by heating at 92 °C for 2 min followed by slow cooling down to room temperature.

### Protein expression and purification
Human PCNA and RFC were purified using previously published protocols[42,44]. *E. coli* codon-optimized sequence of untagged full-length human Ligase 1 (Lig1) was cloned into a pE-pRSF-1b vector (Merk Millipore) using Gibson assembly named hereafter Lig1_WT. The Lig1 L392A/M393A/L394A and Q390A/R391A/L392A/M393A/L394A mutants were generated by PCR and named hereafter Lig1_LML and Lig1_QRLML. *E. coli* optimized sequence of N-terminus deleted Lig1_WT (residues 233-919) were cloned in the pE-SUMOpro expression vector (Lifesensors) and named hereafter ΔN_Lig1. The Q390A/R391A/L392A/M393A/L394A mutant was also generated in ΔN_Lig1 plasmid by PCR and named hereafter ΔN_QRLML_Lig1. The Lig1_WT, Lig1_LML and Lig1_QRLML mutants were purified using a slightly modified version of the previously published protocol[45]. The Lig1_WT and mutant plasmids were transformed into *E. coli* strain BL21 (DE3) competent cells (Novagen) and grown on agar plates containing 50 μg/ml kanamycin. Several colonies were randomly selected and checked for expression level. Lig1_WT, Lig1_LML and Lig1_QRLML plasmids were over-expressed by growing the transformed cells into 6 l of 2YT media (Teknova) supplemented with kanamycin separately. Cells were incubated at 24 °C till the OD_600 reached 0.8. Protein expression was induced with 0.2 mM isopropyl β-D-thiogalactopyranoside (IPTG) concentration and the cells were incubated further for 19 h at 16 °C. Cells were harvested by centrifugation at 5500 × g for 15 min, then resuspended in lysis buffer [50 mM Tris pH (7.5), 50 mM NaCl, 1 mM DTT, 1% NP-40, 1 mM EDTA, 10% glycerol and EDTA free protease inhibitor cocktail tablet/50 ml (Roche, UK)]. All subsequent steps were performed at 4 °C. Cells were lysed by lysozyme followed by sonication. Cell debris was removed by centrifugation (22,040 × g, 60 min) and the clear supernatant was directly loaded onto a cation exchanger, 100 ml phosphocellulose column (Whatman) pre-equilibrated with buffer A [50 mM Tris pH (7.5), 50 mM NaCl, 1 mM DTT, 1 mM EDTA and 10% glycerol]. After loading the sample, the column was washed with 200 ml of buffer A followed by a 100 ml gradient using buffer B [50 mM Tris pH (7.5), 1 M NaCl, 1 mM DTT, 1 mM EDTA and 10% glycerol]. The elution fractions were pooled, slowly diluted to 50 mM NaCl concentration, and loaded onto an anion exchanger, HiTrap Q HP 5 ml (Cytiva) pre-equilibrated with buffer A. The column was then washed with 50 ml of buffer A followed by an elution gradient with 50 ml of buffer B. Protein fractions were pooled and diluted to 150 mM NaCl concentration. Diluted fractions were then loaded onto HiTrap Blue HP 5 ml (Cytiva) pre-equilibrated with buffer C [50 mM Tris pH (7.5), 150 mM NaCl, 1 mM DTT, 1 mM EDTA, and 10% glycerol] followed by washing with 50 ml of buffer A and elution gradient with 50 ml of buffer B. Fractions that contained Lig1_WT, Lig1_LML and Lig1_QRLML were collected, concentrated and loaded onto HiLoad 16/600 Super-dex 200 pg (GE Healthcare) equilibrated with storage buffer [25 mM Tris (pH 7.5), 150 mM NaCl, 1 mM DTT and 0.1 mM EDTA]. Fractions

containing proteins were collected, concentrated using Amicon centrifugal filter, flash-frozen in liquid nitrogen and stored at −80 °C in small aliquots. The presence of protein in the column fractions is detected by Coomassie blue staining after SDS-PAGE (Invitrogen). ΔN_Lig1 and ΔN_ QRLML_Lig1 plasmids were transformed into *E. coli* strain BL21 (DE3) cells separately and grown at 37 °C in 2YT media supplemented with kanamycin until $OD_{600}$ of 0.8 and then induced with 0.2 mM IPTG and incubated further for 18 h at 16 °C. Cells were harvested by centrifugation and re-suspended in lysis buffer [50 mM Tris pH (7.5), 750 mM NaCl, 20 mM imidazole, 5 mM $\beta$-mercaptoethanol, 1% NP-40, 10% glycerol and EDTA free protease inhibitor cocktail tablet/50 ml (Roche, UK)]. All subsequent steps were performed at 4 °C. Cells were lysed by adding lysozyme followed by sonication. The lysate was cleared by centrifugation and loaded onto HisTrap HP 5 ml (Cytiva) pre-equilibrated with buffer A [50 mM Tris pH (7.5), 750 mM NaCl, 20 mM imidazole, 5 mM $\beta$-mercaptoethanol and 10% glycerol]. After loading, the column was washed with 50 ml of buffer A followed by 50 ml of washing with low salt buffer B [50 mM Tris pH (7.5), 50 mM NaCl, 20 mM imidazole, 5 mM $\beta$-mercaptoethanol and 10% glycerol]. The bound protein was eluted with a 50 ml elution gradient with buffer C [50 mM Tris pH (7.5), 50 mM NaCl, 500 mM imidazole, 5 mM $\beta$-mercaptoethanol and 10% glycerol]. The protein fractions were pooled and loaded directly loaded onto HiTrap Blue HP 5 ml (Cytiva) pre-equilibrated with buffer D [50 mM Tris pH (7.5), 50 mM NaCl, 5 mM $\beta$-mercaptoethanol and 10% glycerol]. Bound fractions were washed with 50 ml of buffer D followed by a 50 ml gradient with buffer E [50 mM Tris pH (7.5), 1000 mM NaCl, 5 mM $\beta$-mercaptoethanol and 10% glycerol]. The elution fractions containing the ΔN_Lig1 and ΔN_ QRLML_Lig1 were pooled and dialyzed overnight in a dialysis buffer [50 mM Tris pH (7.5), 200 mM NaCl, 20 mM imidazole, 5 mM $\beta$-mercaptoethanol and 10% glycerol] in the presence of SUMO protease (LifeSensors) to remove the SUMO tag to generate native ΔN_Lig1 and ΔN_ QRLML_Lig1. The dialyzed fractions were loaded again onto the HisTrap column using buffers B and C as mentioned above and the un-tagged proteins were collected in the flow-through fractions. Fractions that contained ΔN_Lig1 and ΔN_ QRLML_Lig1 were collected, concentrated, and loaded separately onto HiLoad 16/600 Superdex 200 pg (Cytiva) pre-equilibrated with the storage buffer [50 mM Tris pH (7.5), 150 mM NaCl, 1 mM DTT and 10% Glycerol]. Fractions containing protein were concentrated using Amicon centrifugal filters, flash frozen and stored at −80 °C in small aliquots.

Full-length FEN1 D181A mutant in the pE-SUMOpro expression vector (Lifesensors) was transformed into *E. coli* strain BL21 (DE3) cells and grown at 37 °C in 2YT media supplemented with kanamycin until $OD_{600}$ of 1. Cells were induced with 0.3 mM IPTG and incubated further for 18 h at 16 °C. Cells were harvested by centrifugation at 5500 × *g* for 15 min, then re-suspended in lysis buffer [50 mM HEPES pH (7.5), 750 mM NaCl, 40 mM imidazole, 5 mM $\beta$-mercaptoethanol, 0.1% NP-40, 10% glycerol and EDTA free protease inhibitor cocktail tablet/50 ml (Roche, UK)]. All subsequent steps were performed at 4 °C. Cells were lysed enzymatically by adding 2 mg/ml lysozyme and mechanically by sonication. The lysate was cleared by centrifugation and loaded onto HisTrap HP 5 ml (Cytiva) pre-equilibrated with buffer A [50 mM HEPES pH (7.5), 750 mM NaCl, 40 mM imidazole, 5 mM $\beta$-mercaptoethanol and 10% glycerol]. After loading, the column was washed with 50 ml of buffer A followed by a 50 ml elution gradient with buffer B [50 mM HEPES pH (7.5), 500 mM NaCl, 500 mM imidazole, 5 mM $\beta$-mercaptoethanol and 10% glycerol]. The protein fractions were pooled and dialyzed overnight in a dialysis buffer [50 mM HEPES (pH 7.5), 500 mM NaCl, 5 mM β-Mercaptoethanol and 10% Glycerol] in the presence of SUMO protease (LifeSensors) to remove the SUMO tag to generate native FEN1 D181A. The dialyzed fractions were loaded again onto the HisTrap column using the buffers A and B as mentioned above and the un-tagged protein was collected in the flow-through fractions. Fractions that contained FEN1 D181A were collected, concentrated, and loaded onto HiLoad 16/600 Superdex 75 pg (Cytiva) pre-equilibrated with the storage buffer [50 mM HEPES (pH 7.5), 150 mM NaCl, 1 mM DTT and 10% Glycerol]. Fractions containing FEN1 D181A were concentrated using Amicon centrifugal filters, flash frozen and stored at −80 °C in small aliquots.

## Protein labeling
PCNA N107C was labeled with Cy3 or Cy5 maleimide (GE Healthcare) to a final stoichiometry of 2.7:1 Cy5 to PCNA trimer or 3:1 Cy3 to PCNA trimer. For both labeled proteins, the chemical labeling reactions and free-dye removal steps were carried-out identically as described in Kim et al.[46].

## Protein−protein electrophoresis mobility shift assay (EMSA)
EMSA experiments were conducted in a reaction buffer containing 50 mM HEPES- KOH pH 7.5, 5% (v/v) Glycerol, 1 mM Dithiothreitol (DTT), 0.1 mg/ml bovine serum albumin (BSA), 100 mM KCl, 10 mM $MgCl_2$,1 mM ATP and 1 nM PCNA-Cy5 with increasing concentrations of Lig1. Reaction mixtures were incubated at RT for 20 min, then 5% (v/v) Ficoll was added to the reactions and the entire reaction volume was loaded onto 6% non-denaturing TBE-PAGE gels. The gels were run for 1 h at room temperature (RT) at 70 V in TBE buffer. Gels were visualized using Typhoon Trio (GE Healthcare). Gel data were analysed with GelAnalyser v19.1. The percentage of free PCNA was estimated by dividing the intensity of the band corresponding to free PCNA in each lane by the intensity of the band corresponding to the PCNA alone condition. The percentage of bound PCNA was estimated by subtracting the percentage of free PCNA from 100%. Binding isotherms were fitted to parabolic dependencies[47] as:

$$\text{Bound PCNA (\%)} = 100\,\frac{PL(L_0)}{P_0} = \frac{100\left[L_0 + P_0 + K_D - \sqrt{(L_0 + P_0 + K_D)^2 - 4L_0 P_0}\right]}{2P_0}\,(\%) \quad (1)$$

where $L_O$ and $P_O$ denote the total Lig1 and PCNA concentrations respectively, $PL$ denotes the amount of Lig1-bound PCNA and $K_D$ denotes the dissociation constant. $P_O$ was fixed to 1 nM and the $K_D$ was obtained from the fit.

## Steady-state fluorescence retention experiments
Fluorescence emission spectra were measured by using a Fluoromax-4 (HORIBA JOBIN YVON) spectrofluorometer. Samples were excited at 480 nm and emission was collected from 550 to 750 nm, with an increment of 1 nm and an integration time of 0.2 s. Both emission and excitation slits were set to 5 nm and a 550 nm cut-off filter was placed on the emission side to prevent excitation light leakage into the emission pathway. The temperature was maintained at 22 °C. Emission and excitation polarizers were set to 0° and 54.7° respectively (VM configuration) to eliminate polarization anisotropy effects. FRET spectra were blank-corrected and normalized identically as described in Raducanu et al.[48]. Direct excitation (at 480 nm) single-color spectra were normalized by the total emission intensity of a direct sum of the donor (Cy3) and acceptor spectra (Alexa Fluor 647). For the normalized direct-summed spectrum and for the normalized FRET spectra, the experimental spectrum data points were fitted with a linear combination of Cy3 and Alexa Fluor 647 spectra. The coefficient of the Alexa Fluor 647 contribution to each emission spectrum was recorded. We could not calculate an exact FRET efficiency due to PCNA labeling stoichiometry.

## Lig1 multiple-turnover kinetics assays
Lig1 multiple-turnover assays were performed in a buffer containing 50 mM HEPES-KOH pH 7.5, 5% (v/v) Glycerol, 1 mM Dithiothreitol (DTT), 0.1 mg/ml bovine serum albumin (BSA), 100 mM KCl, 10 mM $MgCl_2$ and 1 mM ATP. Prior to Lig1 addition, 500 nM nick DNA

(biotinylated at both termini) was pre-incubated with 1 µM NeutrAvidin, 500 nM PCNA, 500 nM RFC and 500 nM FEN1 D181A at 37 °C for 1 min. Reactions were initiated by Lig1 (1 nM) addition and further incubated for the indicated amount of time at 37 °C. The reactions were quenched by the addition of 40 mM EDTA. All reactions were incubated with Proteinase K at 50 °C for 15 min and stopped by adding an equal volume of stop buffer (50 mM EDTA, 95% Formamide). DNA in the quenched reactions was denatured by heating at 95 °C for 5 min and then immediately placed on ice. DNA reaction products were separated on 20% denaturing Urea-PAGE gels and visualized using Typhoon Trio (GE Healthcare). Gel data were analysed with GelAnalyser v19.1. Products were quantified as a percentage of product intensity divided by total lane intensity. Product percentages were then converted to product amounts by multiplication with the initial substrate concentration. Product formation curves were fitted to single-exponential burst equations[49] as:

$$P(t) = A\left(1 - e^{-t/\tau_{obs}}\right) \qquad (2)$$

where $A$ is an amplitude constant (in nM) and $\tau_{obs}$ is the inverse of the apparent product formation rate. We mention that these two parameters do not have any physical meaning as burst equations are not an accurate description for multiple-turnover kinetics. An exact equation for fitting can be found in Schnell et al.[50]. Nevertheless, we employ this burst fitting only to estimate the initial rate of the reaction, which can offer improved results compared with a linear fitting while still maintaining simplicity as compared to the exact equation. To estimate the normalized initial reaction rate, we take the derivative of the burst with respect to time near $t = 0$ and divide it by Lig1 (1 nM) concentration as:

$$\frac{v_0}{[E_0]} = \frac{1}{[E_0]}\frac{d}{dt}P(t)\Big|_{t=0} = \frac{1}{[E_0]}\frac{d}{dt}\left[A\left(1 - e^{-t/\tau_{obs}}\right)\right]\Big|_{t=0} = \frac{A}{\tau_{obs}[E_0]} \qquad (3)$$

### Cryo-EM grid preparation and data collection

For each of the datasets of the Lig1−DNA−PCNA complex, a 50 µl inject containing 4 µM DNA nick substrate, 4 µM PCNA trimer and 4 µM Lig1 in a buffer comprising of 25 mM HEPES (pH 7.5), 100 mM K-Ac, 10 mM MgCl₂, 0.5 mM TCEP, with or without 0.1 mM ATP, was loaded onto a Superdex 200 increase 3.2/300 column (GE Life Sciences) equilibrated with a buffer comprising 25 mM HEPES (pH7.5), 100 mM K-Ac, 10 mM MgCl₂ and 0.5 mM TCEP. 3 µl of the eluted fractions were used for grid preparation. For the Lig1−DNA−PCNA−FEN1 toolbelt dataset, the complex was reconstituted in a buffer comprising 25 mM HEPES (pH 7.5), 100 mM K-Ac, 10 mM MgCl₂, 0.5 mM TCEP and 0.1 mM ATP at 4 °C. A 50 µl inject containing 4 µM DNA nick substrate, 4 µM PCNA trimer, 4 µM Lig1 and 4 µM FEN1 was loaded onto a Superdex 200 increase 3.2/300 column (GE Life Sciences) equilibrated with a buffer comprising 25 mM HEPES (pH7.5), 100 mM K-Ac, 10 mM MgCl₂ and 0.5 mM TCEP. CHAPSO (8 mM) was added before 3 µl of the eluted fraction was used for grid preparation. For all complexes, UltrAuFoil® R1.2/1.3 Au grids were glow discharged for 5 min at 40 mA on a Quorum Gloqube glow-discharge unit, then covered with a layer of graphene oxide (Sigma) prior to application of the sample. The sample was blotted for 3 s and plunge frozen into liquid ethane using a Vitrobot Mark IV (FEI Thermo Fisher), set to 4 °C and 100% humidity. Data were collected on a Thermo Fisher Scientific Titan Krios G3 transmission electron microscope with a K3 direct electron detector (Gatan Inc.) at the Midlands Regional Cryo-EM Facility at the Leicester Institute of Structural and Chemical Biology. Data for the Lig1−DNA−PCNA complex reconstituted without ATP were collected in super resolution mode with a calibrated pixel size of 0.835 Å and a dose rate of 18 e⁻/pix/s. Data for

the Lig1−DNA−PCNA complex reconstituted with ATP were collected in super resolution mode with a calibrated pixel size of 1.086 Å and a dose rate of 16.5 e-/pix/s. Data for the Lig1−DNA−PCNA−FEN1 toolbelt were collected in counted mode or super resolution mode with a calibrated pixel size of 0.835 Å and a dose rate of 18 e-/pix/s. The data were collected with EPU 2.12 and acquired using a defocus range between −2.5 and −0.8 µm, for the Lig1−DNA−PCNA complex datasets. For the Lig1−DNA−PCNA−FEN1 toolbelt dataset, data were acquired using a defocus range between −2.5 and −1.0 µm, in 0.3 µm intervals. For the FEN1−PCNA−DNA complex, a 50 µl inject containing 4 µM DNA nick substrate, 4 µM PCNA trimer and 4 µM FEN1 (D181A) in a buffer comprising of 25 mM HEPES (pH 7.5), 40 mM K-Ac, 10 mM MgCl₂, 0.5 mM TCEP, with 0.1 mM ATP, was loaded onto a Superdex 200 increase 3.2/300 column (GE Life Sciences) equilibrated with a buffer comprising 25 mM HEPES (pH7.5), 40 mM K-Ac, 10 mM MgCl₂ and 0.5 mM TCEP. 3 µl of the eluted fractions were used for grid preparation. Data for the FEN1−PCNA−DNA complex were collected in super resolution mode with a calibrated pixel size of 0.835 Å and a dose rate of 18 e-/pix/s. The data were collected with EPU 2.12 and acquired using a defocus range between −2.5 and −1.0 µm, in 0.3 µm intervals.

### Cryo-EM data processing

Preprocessing of all datasets was performed in RELION-3.1[51]. For the Lig1−DNA−PCNA complexes, imported movie stacks were corrected for beam-induced motion and then integrated using MotionCor2[52]. All frames were retained and a patch alignment of 5 × 5 was used. Contrast transfer function (CTF) parameters for each micrograph were estimated by CTFFIND-4.1[53]. Integrated movies were inspected with RELION-3.1 for further processing. Particle picking was performed in an automated mode using crYOLO[54]. All further processing was performed in RELION-3.1. Particle extraction was carried out using a box size of 258 pixels (pixel size: 0.835 or 1.086 Å/pixel). For the Lig1−DNA−PCNA complex reconstituted without ATP, the initial dataset consisted of 2941 movies. 415322 particles were extracted and cleaned by 2D classification followed by 3D classification with alignment. 3D refinement with 2 rounds of polishing and 1 round of per-particle CTF refinement yielded a 4.58 Å map comprising 73886 particles. The final half-maps of this reconstruction were used to produce a density modified map using the Phenix's tool ResolveCryoEM[20]. For the Lig1−DNA−PCNA complex reconstitued with ATP, the initial dataset consisted of 2540 movies. Extracted particles were cleaned by 2D and 3D classification with alignment, yieldying two main 3D classes (Class 1 and 2, Supplementary Fig. 6). Class 1 showed directional anisotropy and was not processed further. 3D refinement of Class 2, and several rounds of polishing and per-particle CTF refinement yielded a 4.19 Å map comprising 107,550 particles and was termed the *open conformation*. For the Lig1−DNA−PCNA−FEN1 complex, the initial dataset consisted of 3302 and 4345 movies collected with the speciment stage tilted by 0° and 30°, respectively. Particles from the untilted and tilted datasets were processed separately. 938654 extracted particles from the untilted dataset were cleaned by 2D classification followed by 5 rounds of 3D classification with alignment. 3D refinement and one round of polishing yielded a 3.24 Å map comprising 272,160 particles, presenting moderate directional anisotropy. 965,550 extracted particles from the tilted dataset were cleaned by 2D classification followed by four rounds of 3D classification with alignment. 3D refinement and one round of polishing and per-particle CTF refinement yielded a 5.52 Å map comprising 104,911 particles. After the removal of particles of over-represented views, particles of the untilted and tilted datasets were combined and subjected to 3D refinement post

processing in RELION 3.1[24] to yield a final 4.40 Å map showing only minor anisotropy. To probe the mobility of Lig1 and DNA relative to PCNA in the Lig1−DNA−PCNA and Lig1−DNA−PCNA−FEN1 complexes, multi-body refinement was performed with RELION 3.1[24]. The complexes were divided into two discrete bodies composed of Lig1−DNA as the first body and PCNA or PCNA−FEN1 as the second body. Soft masks for multi-body refinement were made in RELION 3.1 from the consensus map. The maps for the two discrete bodies after multi-body refinement were post-processed individually and combined. Normal mode analysis of motion was performed in RELION 3.1[51]. For the FEN1−DNA−PCNA−complex, the initial dataset consisted of 3552 movies collected with the speciment stage tilted by 30° to improve the angular distribution of particles. Preprocessing was performed as above. Particle extraction was carried out using a box size of 216 pixels (pixel size: 0.835 Å). Particles were cleaned by 2D classification followed by two rounds of 3D classification with alignment. 3D refinement and solvent masking yielded a 7.8 Å map comprising 72,185 particles. The final half-maps were then used to produce a sharpened map with anisotropy correction using the local_aniso_sharpen tool in Phenix[55].

## Heterogeneous reconstruction by deep neural networks

CryoDRGN v1[23] models were trained on the 107,550 particles obtained from the refined 3D structure of the open conformer of the Lig1−DNA−PCNA complex (i.e. using Bayesian polished and CTF refined particle images). The refined class was deemed to be compositionally homogeneous. Particle orientations and translations (poses) and CTFs were parsed from their assignments as part of the above the 3D refinement. The original image dimensions (220 × 220 pixels at 1.086 Å/pixel) were intially downsampled by Fourier cropping to 104 × 104 pixels (2.297 Å/pixel), in order to optimally benefit from faster mixed-precision training (by using a multiple of 8 box size). The images were then trained for 50 epochs using a 256 × 3 (nodes per layer × layers) architecture for both the encoder and decoder networks. The latent space dimensionality |z| was 8. Any junk images were filtered out by interactively selecting out outliers in the UMAP projection of the latent embeddings, including over one subsequent training using a 1024 × 3 model for 100 epochs. The kept particles were then retrained using a 256 × 3 architecture for 50 epochs as this was sufficient for learning OBD flexibility. Representative density maps were obtained by partitioning the latent encodings into 20 regions via k-means clustering; with the volumes then generated from data points closest in Euclidean distance to the cluster centers.

## Molecular modeling

*Model building of the Lig1−DNA−PCNA complex reconstituted without ATP:* The X-ray structure of the human Lig1−DNA−AMP complex (PDB ID 1X9N)[9], and the structure of the PCNA homotrimer (from PDB ID 1AXC)[56] were rigid-body fitted into the cryo-EM map using Chimera[57]. The upstream 8 base pairs of B-form duplex DNA threading PCNA were built with Chimera[57] and Coot[58], and real-space refined with Coot. The Lig1 DBD loop spanning residues 385-392 containing PIP$_{DBD}$ was built and refined with Coot. The AMP ligand was modeled based on the X-ray structure[9]. The entire model of the complex was subjected to real-space refinement in Phenix[55] with the application of secondary structure and stereochemical constraints. *Model building of Lig1−DNA−PCNA complex in open conformation:* The X-ray structure of the human Lig1−DNA−AMP complex (PDB ID 1X9N)[9], and the structure of PCNA (from PDB ID 1AXC)[56] were rigid-body fitted into the map. The OBD region and AMP were deleted from the model due to lack of the corresponding density. The poor definition of the DNA and Lig1 AdD portions of the map prevented a definitive modeling of these regions, and therefore only the coordinates of the DBD and PCNA were real-

spaced refined and deposited in the PDB. *Model building of the Lig1−FEN1−DNA−PCNA toolbelt:* The Lig1−DNA portion of the cryo-EM model obtained from reconstitution without ATP, and the X-ray models of PCNA and FEN1 (chain Y) from PDB ID 1UL1[32] were individually rigid-body fitted into the cryo-EM map with Chimera[57]. Flexible regions corresponding to the Cap/Helical gateway of FEN1 were deleted from the model. FEN1 hinge (residues 333−336) was edited with Coot to have the C-terminal PIP-box anchored to its binding site on PCNA. The entire model of the complex was subjected to real-space refinement in Phenix[55] with the application of secondary structure and stereochemical constraints. Steric clashes in the final models were alleviated using Isolde[58] and validated using Phenix[55] (Supplementary Table 1 and Supplementary Fig. 15). For modeling of the FEN1−DNA−PCNA complex, the structure of FEN1 bound to product nicked DNA (PDB ID 3Q8K[28]) and PCNA were rigid-body fitted into the cryo-EM map. FEN1 hinge (residues 333−336) was edited with Coot to have the C-terminal PIP-box anchored to its binding site on PCNA. The upstream DNA of B-form duplex DNA threading PCNA were built with Chimera[57] and Coot[59].

## AlphaFold prediction and MD simulations

The AlphaFold model was built with ColabFold 1.3.0[60] generating five models with three recycles. The best ranking model had a pLDDT of 82.2 and a ptmscore of 0.769. The model was built using AlphaFold2-multimer-v2[61] and MMseq2[62]. The MD simulations to explore the OBD dynamics were prepared in the following steps. We started from the cryo-EM model of the Lig1−DNA−PCNA complex, and solvated the system with a truncated dodecahedron box at least 1.2 nm from the protein or DNA atoms. Sodium and chloride ions were added to produce a final concentration of 150 mM of this salt in a neutralized system. The system was minimized and thermalized with two consecutive NVT and NPT simulations of 2 ns each with restraints in the biomolecular atoms. Then 60 ns of unrestrained NPT equilibration were run before pulling the center of mass of the OBD away from the center of mass of the 10 base pairs of DNA that it was bound to. The pulling rate was set to 0.005 nm ps$^{-1}$ and the harmonic constant to 1000 kJ mol$^{-1}$ nm$^{-2}$. Finally, 100 ns with the restrained kept at 5.5 nm with a constant of 500 kJ mol$^{-1}$ nm$^{-2}$ were used to generate five structures in Supplementary Fig. 14. Namely, after coordinate superposition, the trajectory was projected onto its first two Cartesian principal components, and the five points with the largest distance among them were selected. All simulations were run with Charmm36 force field (July 2021 update)[63,64] in Gromacs 2020.6[65–67]. We used MDTraj[68] and scikit-learn[69] to perform trajectory analysis, including the principal component analysis.

## Reporting summary

Further information on research design is available in the Nature Portfolio Reporting Summary linked to this article.

## Data availability

The data that support this study are available from the corresponding authors upon request. The map of the Lig1−DNA−PCNA complex reconstituted without ATP has been deposited in the EMBD with accession code EMD-14078, and the atomic model in the Protein Data Bank under accession code 7QNZ. The consensus and multi-body refined maps of the *open* conformation of the Lig1−DNA−PCNA complex reconstituted with ATP have been deposited in the EMBD with accession code EMD-15921, and the atomic model in the Protein Data Bank under accession code 8B8T. The map of the FEN1−DNA−PCNA complex has been deposited with accession code EMD-15385. The map of the Lig1−DNA−PCNA−FEN1 complex has been deposited in the EMBD with accession code EMD-14080, and the atomic model in the

Protein Data Bank under accession code 7QO1. Source data are provided with this paper.

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

## Acknowledgements

This research was supported by King Abdullah University of Science and Technology through core funding (to S.M.H. and A.D.B.) and the Competitive Research Award Grant CRG8 URF/1/4036-01-01 (to S.M.H. and A.D.B.). We acknowledge The Midlands Regional Cryo-EM Facility at the Leicester Institute of Structural and Chemical Biology (LISCB), major funding from MRC (MC_PC_17136). We thank Christos Savva and T.J. Ragan (LISCB, University of Leicester) for their help in cryo-EM data collection and advice on data processing. This project has been carried out using the resources of CSUC.

## Author contributions

M.T. purified all the proteins and their variants, with the help of F.R.; M.T. and V.S.R. carried out the activity and binding assays; K.B. prepared the cryo-EM samples and acquired the data; K.B., T.S., and A.D.B. analyzed the cryo-EM data; T.S. performed the cryo-EM particle heterogeneity analysis. K.B., C.L., and A.D.B. built the molecular models. R.C. performed the AlphaFold prediction and MD simulation. S.M.H. and A.D.B. conceived the research and wrote the article. All authors discussed the results and commented on the manuscript.

## Competing interests

The authors declare no competing interests.
