## [Peer Review File · Nature Communications]

Mechanism of human Lig1 regulation by PCNA in Okazaki fragment sealingReviewers' Comments:

Reviewer #1:

Remarks to the Author:

The authors present cryo-EM structures of a Lig1-DNA-PCNA complex in the absence or presence of ATP. In addition is a structure of a FEN1-Lig1-DNA-PCNA complex presented, analogous to a previously, by them, published cryo-EM structure of a PolD-FEN1-DNA-PCNA complex. The new structures are important, but is in the present form only providing incremental new insights in the function of Lig1, and the coordinated action between POLD, FEN1 and Lig1 during the Okazaki-fragment maturation process. The observed physical interaction between Lig1 and PCNA does identify a non-canonical PCNA interaction motif called PIPDBD . Unfortunately, substituting the three residues that should be critical for the interaction with PCNA only provide a partial loss of interaction with PCNA (Figure 3). Thus, the partial loss of interaction does limit the conclusions that can be drawn from the remaining experiments. On a final note, the model presented in Figure 5 does not account for how Pol delta repeatedly hands off 5´-flaps that are 1-2 nts to FEN1. FEN1 preferentially removes 5´-flaps that are 1-2 nts . After cleavage, Pol delta creates a new flap that is 1-2 nts. This process continues repeatedly in 1-2 nts long steps until all RNA is removed and Lig1 can seal the nick. As pointed out in Figure 5, Pol delta and Lig1 are competing for the same position on PCNA, and the authors suggest that Pol delta will create a single flap, then hand it over to FEN1 and after that be released from PCNA. Next, FEN1 cleaves the flap and hands it over to Lig1 that has bound to PCNA where Pol delta was previously bound. This only describes one cycle, whereas the Okazaki-maturation process requires multiple cycles. Thus, the presented model in Figure 5 does not agree with the detailed biochemical characterization of how Pol delta, FEN1 and Lig1 are coordinated during Okazaki-fragment maturation (See for example, Garg et al Genes&Dev 2004 18:2764-2773, and Stodola and Burgers, Nature Struct & Mol Biol. 2016 23:402-408).

Some specific points

Line 27, in the abstract, The authors write "During lagging strand synthesis, Lig1, cooperates with the sliding clamp PCNA to seal the nicks between Okazaki fragments generated by FEN1...". This should be rephrased since two separate mechanisms, idling by Pol delta and cleavage of the 5´-flap by FEN1, generate nicks that can be sealed by Lig1.

Line 45, "contiguous" should it be "continuous"?

Line 46, Pol alpha is described as a low fidelity DNA polymerase. It is unclear why the authors want to make that point in this manuscript. Furthermore, the accuracy of Pol alpha is high considering that Pol alpha lacks an exonuclease activity that removes errors. In comparison, exonuclease deficient Pol epsilon has a higher error rate than Pol alpha, and other polymerases lacking exonuclease activity has also a much higher error-rate.

Reviewer #2:

Remarks to the Author:

The manuscript by Blair et al. presents four cryo-EM structures of macromolecular complexes consisting of Lig1, PCNA, Fen1, DNA containing a break in a phosphodiester backbone and excess ATP, mixed in different combinations, namely: Lig1-DNA-PCNA-no ATP, Lig1-DNA-PCNA-ATP in "closed" conformation, Lig1-DNA-PCNA-ATP in "open" conformation, and Lig1-DNA-PCNA-Fen1-no ATP. The structural data are accompanied by FRET analysis, EMSA-binding and ligation activity assays. The current study follows on manuscripts published previously by the same groups reporting cryo-EM structures of similar multiprotein assemblies, i.e. Pol delta-DNA-PCNA and Pol delta-DNA-PCNA—Fen1 (Nature Commun., 2020, 11:1109) and Pol kappa-DNA-PCNA (Nature Commun., 2021, 12:6095), and complements previously published X-ray structures of Lig1-DNA (Nature 2004 432:473), FEN1-PCNA (EMBO, 24:683), and SAXS model of Lig1-PCNA (Mol. Cell 2006, 24:279). For the first time, authors unveil both, FEN1 and Lig1 simultaneously binding to PCNA via their PIP-box motifs. Based on their

findings, authors reiterate a “toolbelt model”, in which proteins involved in processing of Okazaki fragments successively assemble on PCNA scaffold.

Here are my comments and suggestions concerned with sample preparation and interpretation of the maps:

-In regard to data sets 1 and 2 summarized in Supplementary Fig. 7 representing Lig1-DNA-PCNA-ATP “closed” and Lig1-DNA-PCNA-ATP “open”, respectively. Was the sample prep for these two data sets different? If not, I would expect a mixture of the two conformers in both data sets. Can you comment on this issue?

-Structures of Lig1-DNA-PCNA-no ATP and Lig1-DNA-PCNA-ATP in “closed” conformation seem structurally analogous, yet the Lig-1-OBD is not visible in the latter structure. Can the authors speculate on this topic? Does the addition of excess ATP influence the mobility of OBD? A similar observation applies to a structure of Lig1-DNA-PCNA-ATP in “open” conformation, where the OBD is not visible.

-On the same note, do the authors observe the ATP density in the active site of Lig1 in presented structures?
Are the active site residues located in close proximity to the nick in DNA phosphodiester backbone?
Please expand discussion in the manuscript.

-Please also expand on the role of the N-terminal PIP box from Lig1 in mediating Lig1-PCNA interaction. Does the N-terminal domain of Lig1 essentially behave like an intrinsically-disordered protein? Would the authors expect the domain to be disordered in, for instance, Lig1-PCNA complex lacking the DNA? Is there any structural evidence addressing the function of the N-terminal PIP from Lig1? Authors speculate (lines 187-189) that the structural data argue the N-terminal PIP motif facilitates initial recruitment of Lig1 by PCNA? How the data support this notion if N-terminal PIP is not visible in the structures? Authors might consider supplementing the manuscript with, for example, FRET or binding data showing the interaction of PCNA +/- DNA and a mutant Lig1 harboring the N-terminal PIP truncation. If such a study has already been performed, please include relevant references.

-In Lig1-DNA-PCNA-Fen1-no ATP structure, Lig1 lacks any direct contacts with Fen1. What is the structural basis for Fen1 enhancement of Lig1 activity (Fig 4a)? If, as authors suggest (line 256), Lig1-OBD plays a role in modulating this enhancement, please provide supporting structural evidence. For instance, are the position/positions of OBDs in Lig1-DNA-PCNA-Fen1-no ATP and Lig1-DNA-PCNA-no ATP structures different? I’m curious about the DNA-binding affinities of Lig1-DNA-PCNA-no ATP and Lig1-DNA-PCNA-Fen1-no ATP. Are the Kds different? Please expand.

-Is the DNA in Lig1-DNA-PCNA-Fen1-no ATP adenylated?

Some technical comments:

-Can the authors speculate on the location of Cy3 labels in PCNA? Location of labels would affect the outcome of FRET experiments.

-Line 517: please elaborate on “moderate directional anisotropy”. Can the authors provide, for instance, a 3D-FSC?

-Line 521: “Removal of particles of overrepresented views” improved the directional resolution anisotropy. I find this statement surprising, as an overall loss of resolution along the axis parallel to the preferred orientation axis is related to the absence of certain views, rather than overrepresentation of others. In other words, I wouldn’t expect an increase in quality and resolution of

a 3D map after removing some 2D class. Contrary, the quality/resolution of the reconstruction might decrease.

-All structures seem to suffer from the preferred orientations, as evidenced by angular distribution plots in Supplementary Fig. 2, 6 and 12. As for Supplementary Fig. 12, the preferred orientation problem seems to be still prevalent despite of tilting. Please include representative sets of 2D class averages demonstrating the presence of different 2D projections of the complex. The most reliable way to confirm the preferred orientation problem has been overcome would be an inspection of 3D maps, for example, in Chimera. Therefore, I would appreciate the opportunity to look at these maps. Maps are currently not available for viewing in the EMDB. Likewise, I would also appreciate the opportunity to inspect models derived from the maps.

-I noticed a strange dip in FSC curves at ~6Å presented in Supplementary Figures 2c & e, 6c, e & h, and 12g. Could the authors speculate on this issue? Could this be caused by misalignments of particles?

Minor corrections:

-Figure legend to Supplementary Fig. 2: first "f)" should be replaced with "g" in this caption.

-Supplementary Table 1: I think authors can find B-factor values in the output from Phenix Resolve cryo-EM and include them in the table.

Reviewer #3:

Remarks to the Author:

The manuscript entitled "Mechanism of human Lig1 regulation by PCNA in Okazaki (OK) fragment sealing," by Blair et al presents a structural characterization of OK fragment maturation. The authors solved several cryo-EM structures of human Lig1 bound to nicked DNA and PCNA in the absence and presence of FEN1. With the structural information and biochemical analyses, the authors proposed a model in which the PCNA clamp serves as a platform for Lig1 to cooperate with FEN1 for OK fragment processing through a substrate-handover process. Technically, the structural data presented here is sound, which is important for understanding the maturation of OK fragment. However, at present the manuscript is lack of important biochemical and structural analyses to provide mechanistic insights into OK fragment processing by FEN1 and Lig1. If the concerns listed below can be satisfactorily addressed, the manuscript would be appropriate for publication in Nature Communications.

Major points:

1 The structure of an archaeal Lig1-PCNA-DNA complex has been recently reported (Sverzhinsky et al 2022). This archaeal structure is very similar to its human counterpart presented in this manuscript. The author should compare these two structures to reveal if OK fragment processing is conserved across species.

2 In the human Lig1-PCNA-DNA structure, the PIP loop of Lig1 is the only motif engaging with PCNA. However, the PIP loop mutant, Lig1-LML, still retains a Lig1-PCNA interaction comparable to the Lig1 WT even in the presence of substrate DNA. It is not clear how the PIP loop contributes to Lig1 docking onto PCNA and nicked DNA. Additional mutant(s), such as removal of the entire PIP loop from the Lig1-DBD domain, should be generated to further characterize the importance of the Lig1-PCNA interaction in processing OK fragment in the absence and presence of FEN1.

3 With the result shown in Supplementary Fig. 10, the authors claimed that PCNA does not stimulate the ligation activity of human Lig1. This is in contrast to the observation in archaea (Sverzhinsky et al 2022). Please could the authors provide explanation to this difference between archaea and human?

The authors need to provide more compelling biochemical evidence to rule out the role of PCNA in stimulating Lig1 activity. Please refer to the in vitro assays performed by Sverzhinsky et al.

4 In the Lig1-FEN1-PCNA structure, FEN1 assumes an upright conformation through interactions with PCNA and Lig1-OBD. However, Lig1-OBD appears mobile relative to FEN1 in the toolbelt. Based on this observation, the authors proposed that Lig1-OBD flexibility contributes to a step-by-step handoff of substrate from FEN1-PCNA. This structural feature is the most important finding of this study. It should be further explored. The authors could remove the OBD from Lig1 (Lig1 Δ OBD) to reconstitute the Lig1 Δ OBD-FEN1-PCNA-DNA complex for structural determination. This analysis will help to elucidate the mechanism of how Lig1 cooperate with FEN1 to handoff the nicked DNA substrate on the PCNA clamp.

5. The authors claimed that the interaction between FEN1 and Lig1-OBD contributes to restricting the conformational space of the OBD while this interaction is not direct. This claim should be justified by providing additional evidence. Notably, in the absence of FEN1, the OBD of Lig1 is already located close to the nicked DNA. The authors need to compare between Lig1-PCNA and Lig1-FEN1-PCNA the ratio of the particles showing a stable OBD engaging with the nicked DNA. As FEN1 binds to PCNA through a PIP loop, the authors could consider to generate FEN1 Δ PIP in which the PIP loop is truncated and test if the FEN1 Δ PIP could interact with Lig1 or not.

6. If FEN1 could facilitate the OBD binding to the nicked DNA, the ligation activity of Lig1 should be boosted by FEN1. This could be tested by in vitro ligation assay.

Minor points:

1 The archaeal structure (Sverzhinsky et al 2022) should be appropriately introduced and referenced.

2 Line 127: citations should be given for the neural network algorithm in the main text.

3 Line 158: It is not clear how "20% of the flexibility" was obtained. The relevant information was not provided in Supplementary Fig. 6.

4 Line 194: The mentioned DNA features should be labeled clearly in the relevant figure.

5 Line 504: The detailed strategy for separating the open and close conformations should be provided.

6 In Figure 5, the published structures should be labeled with their corresponding EMD codes.

REVIEWER COMMENTS

Reviewer #1 (Remarks to the Author):

The authors present cryo-EM structures of a Lig1-DNA-PCNA complex in the absence or presence of ATP. In addition is a structure of a FEN1-Lig1-DNA-PCNA complex presented, analogous to a previously, by them, published cryo-EM structure of a PolD-FEN1-DNA-PCNA complex. The new structures are important, but is in the present form only providing incremental new insights in the function of Lig1, and the coordinated action between POLD, FEN1 and Lig1 during the Okazaki-fragment maturation process. The observed physical interaction between Lig1 and PCNA does identify a non-canonical PCNA interaction motif called PIPDBD . Unfortunately, substituting the three residues that should be critical for the interaction with PCNA only provide a partial loss of interaction with PCNA (Figure 3). Thus, the partial loss of interaction does limit the conclusions that can be drawn from the remaining experiments. On a final note, the model presented in Figure 5 does not account for how Pol delta repeatedly hands off 5'-flaps that are 1-2 nts to FEN1. FEN1 preferentially removes 5'-flaps that are 1-2 nts . After cleavage, Pol delta creates a new flap that is 1-2 nts. This process continues repeatedly in 1-2 nts long steps until all RNA is removed and Lig1 can seal the nick. As pointed out in Figure 5, Pol delta and Lig1 are competing for the same position on PCNA, and the authors suggest that Pol delta will create a single flap, then hand it over to FEN1 and after that be released from PCNA. Next, FEN1 cleaves the flap and hands it over to Lig1 that has bound to PCNA where Pol delta was previously bound. This only describes one cycle, whereas the Okazaki-maturation process requires multiple cycles. Thus, the presented model in Figure 5 does not agree with the detailed biochemical characterization of how Pol delta, FEN1 and Lig1 are coordinated during Okazaki-fragment maturation (See for example, Garg et al Genes&Dev 2004 18:2764-2773, and Stodola and Burgers, Nature Struct & Mol Biol. 2016 23:402-408).

We thank the reviewer for acknowledging that our new structures are important. In the revised manuscript, we provide substantial new experimental data that comprehensively address the mechanism of Lig1 recruitment to PCNA. The new FRET data using an array of Lig variants (Figure 3), combined with the cryo-EM structure (Figure 1), support that Lig1 initially recruits PCNA via a high-affinity PIP motif located at the N-terminus. Once Lig1 and PCNA assemble as two rings encircling DNA, Lig1 is tethered to PCNA via the PIP in the DBD, and the N-terminal PIP is released. Ligation assays using the new Lig1 variants support that PIP_{DBD} is critical for substrate handoff from FEN1, while the N-terminal PIP is dispensable (Figure 4). We also present an intermediate resolution cryo-EM structure of FEN1 bound to PCNA and nicked DNA (Figure 5), showing that FEN1 binds one of the three PCNA protomers and grips the DNA sharply bent at the nick in an exposed position above the front face of the clamp, accessible to the incoming Lig1.

Finally, following the referee's suggestions, we modified Figure 5 of the previous manuscript (Figure 6 in the current manuscript), which now shows a section describing the multiple cycles of nick translation. In fact, recent work from our labs, that is currently

under review in *Nature Communications*, supports these multiple cycles of nick translation reactions that removes the primer 1-2 nucleotide at a time in Human, and therefore we feel confident about including it in the model presented in Figure 6.

Some specific points

Line 27, in the abstract, The authors write “During lagging strand synthesis, Lig1, cooperates with the sliding clamp PCNA to seal the nicks between Okazaki fragments generated by FEN1...”. This should be rephrased since two separate mechanisms, idling by Pol delta and cleavage of the 5`-flap by FEN1, generate nicks that can be sealed by Lig1.

For space limitations, it is not possible to describe the full Okazaki fragment maturation mechanism in the abstract. However, the abstract phrase now mentions Pol delta: “...generated by Pol delta and Flap endonuclease 1 (FEN1)”

Line 45, “contiguous” should it be “continuous”?

This has been corrected, thank you.

Line 46, Pol alpha is described as a low fidelity DNA polymerase. It is unclear why the authors want to make that point in this manuscript. Furthermore, the accuracy of Pol alpha is high considering that Pol alpha lacks an exonuclease activity that removes errors. In comparison, exonuclease deficient Pol epsilon has a higher error rate than Pol alpha, and other polymerases lacking exonuclease activity has also a much higher error-rate.

We thank the referee for pointing this out, therefore we removed “low fidelity” when describing Pol alpha activity.

Reviewer #2 (Remarks to the Author):

The manuscript by Blair et al. presents four cryo-EM structures of macromolecular complexes consisting of Lig1, PCNA, Fen1, DNA containing a break in a phosphodiester backbone and excess ATP, mixed in different combinations, namely: Lig1-DNA-PCNA-no ATP, Lig1-DNA-PCNA-ATP in “closed” conformation, Lig1-DNA-PCNA-ATP in “open” conformation, and Lig1-DNA-PCNA-Fen1-no ATP. The structural data are accompanied by FRET analysis, EMSA-binding and ligation activity assays. The current study follows on manuscripts published previously by the same groups reporting cryo-EM structures of similar multiprotein assemblies, i.e. Pol delta-DNA-PCNA and Pol delta-DNA-PCNA—Fen1 (*Nature Commun.*, 2020, 11:1109) and Pol kappa-DNA-PCNA (*Nature Commun.*, 2021, 12:6095), and complements previously published X-ray structures of Lig1-DNA (*Nature* 2004 432:473), FEN1-PCNA (*EMBO*, 24:683), and SAXS model of Lig1-PCNA (*Mol. Cell* 2006, 24:279). For the first time, authors unveil both, FEN1 and Lig1 simultaneously binding to PCNA via their PIP-box motifs. Based on their findings, authors reiterate a “toolbelt model”, in which proteins involved in processing of Okazaki fragments successively assemble on PCNA scaffold.

Here are my comments and suggestions concerned with sample preparation and interpretation of the maps:

-In regard to data sets 1 and 2 summarized in Supplementary Fig. 7 representing Lig1-DNA-PCNA-ATP “closed” and Lig1-DNA-PCNA-ATP “open”, respectively. Was the sample prep for these two data sets different? If not, I would expect a mixture of the two conformers in both data sets. Can you comment on this issue?

The sample preparation that yielded the maps of the two conformers is the same. The sample was separated by micro-SEC prior to grid freezing (Supp Figure 5 in the new manuscript). A single dataset (Dataset 1, Supplementary Figure 7) was acquired which yielded the two 3D classes of the “closed” and “open” conformers. A second dataset (Dataset 2, Supplementary Figure 7) which predominantly contained particles of the “open” conformer, was combined with particles of the first dataset to improve the map of the open conformer. As explained in the text, it is not possible to assign the “open” conformer to a functional state across the nick sealing reaction, and it is not possible to ascertain whether the two conformers can inter-convert between each other. Nonetheless, the two conformers are helpful in making the point that OBD and the DNA can adopt different conformers, and we used this to make some speculation on their functional role during nick product handoff between the Lig and FEN1. We are intending to investigate the conformational states of the Lig and the DNA using FRET and this will be a subject for future study.

-Structures of Lig1-DNA-PCNA-no ATP and Lig1-DNA-PCNA-ATP in “closed” conformation seem structurally analogous, yet the Lig-1-OBD is not visible in the latter structure. Can the authors speculate on this topic? Does the addition of excess ATP influence the mobility of OBD? A similar observation applies to a structure of Lig1-DNA-PCNA-ATP in “open” conformation, where the OBD is not visible.

We thank the referee for this good point. In order to test the reproducibility of the observation of OBD rigidity in the dataset of the Lig1-DNA-PCNA-no ATP complex, we have frozen a new grid of the same sample, acquired a small dataset (910 images), and processed the data to 2D classification (please see figure below).

2D classes support that OBD is mobile in this new dataset. Our conclusion is that there is an intrinsic variability of OBD mobility in the EM experiment, which may depend on small random differences in the sample freezing process, rather than on the presence of ATP in the buffer. Such variability may also depend on the non-ligatable nicked DNA used in this study. We note that different degrees of OBD flexibility were also observed

in recent cryo-EM structures of ssLig bound to PCNA and several DNA substrates (Sverzhinsky, Structure, 2022).

-On the same note, do the authors observe the ATP density in the active site of Lig1 in presented structures?

Are the active site residues located in close proximity to the nick in DNA phosphodiester backbone? Please expand discussion in the manuscript.

We thank the referee for this question. We have carried out an inspection of the active site in the different structures. Inset reporting the map region at the active site of the Lig1-DNA-PCNA-no ATP and Lig1-DNA-PCNA-FEN1 structures are presented in Figure 1d and Supplementary Figure 14j. In these two structures, DNA is adenylated and active site residues are ordered. In the “closed” Lig1-DNA-PCNA structure (Supplementary Figure 8), density protruding from the 5' phosphate at the nick suggests that the DNA is adenylated, but the active site residues are poorly ordered (because AdD is more flexible due to the flexible OBD), while in the “open” structure, the map does not allow to discriminate the adenylation state. Based on this, in the discussion we propose that OBD encirclement of DNA is required to stabilize the active site residues around the nick to promote nick sealing.

-Please also expand on the role of the N-terminal PIP box from Lig1 in mediating Lig1-PCNA interaction. Does the N-terminal domain of Lig1 essentially behave like an intrinsically-disordered protein? Would the authors expect the domain to be disordered in, for instance, Lig1-PCNA complex lacking the DNA? Is there any structural evidence addressing the function of the N-terminal PIP from Lig1? Authors speculate (lines 187-189) that the structural data argue the N-terminal PIP motif facilitates initial recruitment of Lig1 by PCNA? How the data support this notion if N-terminal PIP is not visible in the structures? Authors might consider supplementing the manuscript with, for example, FRET or binding data showing the interaction of PCNA +/- DNA and a mutant Lig1 harboring the N-terminal PIP truncation. If such a study has already been performed, please include relevant references.

Disorder predictions (Figure 1a) and the absence of density in the cryo-EM map (Figure 1c) are consistent with Lig1 N-terminal domain to behave like an intrinsically disordered protein.

We have generated a new array of Lig1 variants to explore the role both the N-terminal region and PIP_{DBD} in PCNA recruitment and used them in the FRET experiment (Figure 3). Collectively, our structural and binding data argue that the interaction mediated by the Lig1 N-terminus facilitates the initial recruitment of PCNA from solution, and that the interaction with the DBD stabilizes the functional complex on nicked DNA. Once Lig1 and PCNA assemble as two-stack rings encircling DNA, PIP_{N-term} is released from PCNA.

It is likely that, before PIP_{DBD} is engaged, PIP_{N-term} tethers Lig1 to PCNA in an ensemble of orientations due to the conformational flexibility of the N-terminal domain. We carried out a structural prediction with AlphaFold (Figure 2), which surprisingly shows an architecture similar to the cryo-EM structure, but with the two main PIPs bound to two

different PCNA monomers. Such prediction may pertain to a state post PIP_{DBD} engagement, and shows that the flexible N-terminus extending from the Lig1 core needs to fold back to stay attached to the second PIP pocket on PCNA: it is possible that the entropic penalty associated with this chain reversion results in the release of PIP_{N-term} once the Lig1 core has assembled around the DNA (Figure 1).

-In Lig1-DNA-PCNA-Fen1-no ATP structure, Lig1 lacks any direct contacts with Fen1. What is the structural basis for Fen1 enhancement of Lig1 activity (Fig 4a)? If, as authors suggest (line 256), Lig1-OBDD plays a role in modulating this enhancement, please provide supporting structural evidence. For instance, are the position/positions of OBDDs in Lig1-DNA-PCNA-Fen1-no ATP and Lig1-DNA-PCNA-no ATP structures different? I'm curious about the DNA-binding affinities of Lig1-DNA-PCNA-no ATP and Lig1-DNA-PCNA-Fen1-no ATP. Are the K_ds different? Please expand.

Our functional data are not consistent with an enhancement of Lig1 activity by FEN1. Rather, FEN1 decreases Lig1 activity when the PIP in the DBD is inactivated (Figure 4). The positions of OBDDs in Lig1-DNA-PCNA-Fen1 and Lig1-DNA-PCNA-no ATP structures are equivalent, while the DNA binding affinities for the two complexes are not known.

We have performed MD simulations probing diverging OBDD conformations suggesting that the space filled by FEN1 does not restrict the conformational space sampled by the OBDD (Supplementary Movie 1; Supplementary Figure 15). We have modified the statement in the discussion accordingly.

-Is the DNA in Lig1-DNA-PCNA-Fen1-no ATP adenylated?

We thank the referee for pointing this out. An inset reporting the map region at the active site of the Lig1-DNA-PCNA-FEN1 structure (obtained from a buffer with ATP) is presented in Supplementary Figure 14j, showing that DNA is adenylated.

Some technical comments:

-Can the authors speculate on the location of Cy3 labels in PCNA? Location of labels would affect the outcome of FRET experiments.

Labeled PCNA carries the well-established N107C mutation (for further information on this mutation please refer, for example, to Figure 1a in <https://www.sciencedirect.com/science/article/pii/S0021967319306077>) in each of the 3 monomers. In the absence of the N107C mutation, at the employed reactive dye:PCNA ratio in the labeling reaction, WT PCNA gets labeled only to unspecific levels of ~0.3 fluorophores / PCNA trimer. When PCNA N107C is employed in the dye coupling reaction the labeling yield is increased to ~3 fluorophores / PCNA trimer and therefore, it is ~10-fold higher than the background level of labeling due to the WT cysteines. Thus, ~90% of the fluorophore molecules are attached to C107 in PCNA N107C. The N107 (C107) residue is located in the back face of PCNA near the subunit contact point. Most importantly, this distant localization of C107 in the back part of

PCNA ensures that the signal change in the attached cyanine dye is not induced by direct photo-modulation of the cyanine dye fluorescence upon PCNA-Lig1 binding, as Lig1 does not interact with the back face of PCNA.

-Line 517: please elaborate on “moderate directional anisotropy”. Can the authors provide, for instance, a 3D-FSC?

3DFSC plots are now presented for all the maps reported in the study (Supplementary Figures 2d, 6e, 6j, 12g and 14e). We do not feel that the residual anisotropy in the presented maps affects any of our conclusions.

-Line 521: “Removal of particles of overrepresented views” improved the directional resolution anisotropy. I find this statement surprising, as an overall loss of resolution along the axis parallel to the preferred orientation axis is related to the absence of certain views, rather than overrepresentation of others. In other words, I wouldn’t expect an increase in quality and resolution of a 3D map after removing some 2D class. Contrary, the quality/resolution of the reconstruction might decrease.

Removal of particles of overrepresented views lowers the overall map resolution (please see the description in the Methods Section) but improves the map appearance because it homogenizes the directional resolution (so that that map does not appear “stretched” along certain angles).

-All structures seem to suffer from the preferred orientations, as evidenced by angular distribution plots in Supplementary Fig. 2, 6 and 12. As for Supplementary Fig. 12, the preferred orientation problem seems to be still prevalent despite of tilting. Please include representative sets of 2D class averages demonstrating the presence of different 2D projections of the complex. The most reliable way to confirm the preferred orientation problem has been overcome would be an inspection of 3D maps, for example, in Chimera. Therefore, I would appreciate the opportunity to look at these maps. Maps are currently not available for viewing in the EMDB. Likewise, I would also appreciate the opportunity to inspect models derived from the maps.

We thank the review for raising this point. All maps of the Lig1 complexes and relative models have been included in the resubmission for the referee’s inspection. We are confident that the referee will agree with us that the residual anisotropy in the presented maps does not affect any of our conclusions.

-I noticed a strange dip in FSC curves at ~6Å presented in Supplementary Figures 2c & e, 6c, e & h, and 12g. Could the authors speculate on this issue? Could this be caused by misalignments of particles?

Such dip is only prominent in the FSC relative of the “open” and “closed” Lig1 conformers (Supplementary Figures 6c and 6i), caused by an early drop in the FSC, likely an effect of the moderate anisotropy associated to these maps. This feature does not seem to be significant for our arguments.

Minor corrections:

-Figure legend to Supplementary Fig. 2: first “f” should be replaced with “g” in this caption.

This has been corrected, thank you.

-Supplementary Table 1: I think authors can find B-factor values in the output from Phenix Resolve cryo-EM and include them in the table.

B-factors have now been included in the table.

Reviewer #3 (Remarks to the Author):

The manuscript entitled “Mechanism of human Lig1 regulation by PCNA in Okazaki (OK) fragment sealing,” by Blair et al presents a structural characterization of OK fragment maturation. The authors solved several cryo-EM structures of human Lig1 bound to nicked DNA and PCNA in the absence and presence of FEN1. With the structural information and biochemical analyses, the authors proposed a model in which the PCNA clamp serves as a platform for Lig1 to cooperate with FEN1 for OK fragment processing through a substrate-handover process. Technically, the structural data presented here is sound, which is important for understanding the maturation of OK fragment. However, at present the manuscript is lack of important biochemical and structural analyses to provide mechanistic insights into OK fragment processing by FEN1 and Lig1. If the concerns listed below can be satisfactorily addressed, the manuscript would be appropriate for publication in Nature Communications.

We thank the reviewer for acknowledging the quality of our work.

Major points:

1 The structure of an archaeal Lig1-PCNA-DNA complex has been recently reported (Sverzhinsky et al 2022). This archaeal structure is very similar to its human counterpart presented in this manuscript. The author should compare these two structures to reveal if OK fragment processing is conserved across species.

We thank the reviewer for pointing out this study that was published during the revision. We now comment on the finding of this study in various places in the manuscript. A comparison with the archaeal structure is now reported in the results section. It appears that the interaction between the DBD loop and PCNA is conserved, despite the divergence of the respective PIP motifs (please see alignments in Figure 1b). However,

the interaction involving the nonPIP region in the DBD, and the interaction involving the AdD observed in the archaeal system do not seem to be conserved in human.

2 In the human Lig1-PCNA-DNA structure, the PIP loop of Lig1 is the only motif engaging with PCNA. However, the PIP loop mutant, Lig1-LML, still retains a Lig1-PCNA interaction comparable to the Lig1 WT even in the presence of substrate DNA. It is not clear how the PIP loop contributes to Lig1 docking onto PCNA and nicked DNA. Additional mutant(s), such as removal of the entire PIP loop from the Lig1-DBD domain, should be generated to further characterize the importance of the Lig1-PCNA interaction in processing OK fragment in the absence and presence of FEN1.

We thank the reviewer for this suggestion. To disambiguate the results of the binding experiments, we have generated a new array of Lig1 variants to probe the role of both the Lig1 N-terminal region and PIP DBD loop in PCNA recruitment, and used them in the FRET experiment (Figure 3). Collectively, our structural and binding data argue that the interaction mediated by the Lig1 N-terminus facilitates the initial recruitment of PCNA from solution, and that the interaction with the DBD stabilizes the functional complex on nicked DNA. Once Lig1 and PCNA assemble as two-stack rings encircling DNA, the N-terminal region containing the high affinity PIP is released from PCNA. Ligation assays using the new Lig1 variants support that PIP_{DBD} is critical for substrate handoff from FEN1, while the N-terminal PIP is dispensable (Figure 4).

3 With the result shown in Supplementary Fig. 10, the authors claimed that PCNA does not stimulate the ligation activity of human Lig1. This is in contrast to the observation in archaea (Sverzhinsky et al 2022). Please could the authors provide explanation to this difference between archaea and human? The authors need to provide more compelling biochemical evidence to rule out the role of PCNA in stimulating Lig1 activity. Please refer to the in vitro assays performed by Sverzhinsky et al.

Because of the topological differences between archaeal and human ligases (in the archaeal ligase the N-terminal domain is absent) as well as the clear differences in the interactions with PCNA (described above), we believe that the effects of PCNA on ligase activity in the two systems cannot be directly compared. Absence of a PCNA-induced stimulation of human Lig1 ligation has been reported by other authors before (see Levin et al, PNAS, 1997; Levin et al., JBC, 2004). This lack of stimulation can be explained if the interaction of Lig1 with PCNA tethers Lig1 to the DNA substrate, but does not significantly improve the ability of Lig1 to locate nicks within the DNA molecule, possibly because the DNA substrates used in the studies are short. However, we do demonstrate that PCNA modulates Lig1 activity once FEN1 is present in the reaction and Lig1 needs to capture the nicked DNA from FEN1 (Figure 4 and Figure 5 in the new manuscript). If such modulation also applies to the archaeal system remains to be determined.

4 In the Lig1-FEN1-PCNA structure, FEN1 assumes an upright conformation through

interactions with PCNA and Lig1-OBD. However, Lig1-OBD appears mobile relative to FEN1 in the toolbelt. Based on this observation, the authors proposed that Lig1-OBD flexibility contributes to a step-by-step handoff of substrate from FEN1-PCNA. This structural feature is the most important finding of this study. It should be further explored. The authors could remove the OBD from Lig1 (Lig1 Δ OBD) to reconstitute the Lig1 Δ OBD-FEN1-PCNA-DNA complex for structural determination. This analysis will help to elucidate the mechanism of how Lig1 cooperate with FEN1 to handoff the nicked DNA substrate on the PCNA clamp.

We deem the referee's interpretation of OBD flexibility interesting, however we did not propose a step-by-step substrate handover mediated by the flexible OBD. OBD flexibility is required because the AdD-DBD needs to be accessible to receive the DNA substrate from FEN1. Our structures of the Lig1-DNA-PCNA complex with a flexible OBD (Supplementary Figure 8) argue that the AdD-DBD is sufficient to stably bind the nicked DNA with different degrees of bending. We now present an intermediate resolution cryo-EM structure of the FEN1-nickedDNA-PCNA complex showing that FEN1 binds one of the three PCNA protomers and grips the DNA sharply bent at the nick in an exposed position above the front face of PCNA (Figure 5). The path of the DNA axis bends $\sim 100^\circ$ at the location of the nick, which is incompatible with an engagement of the OBD. We speculate that Lig1 AdD-DBD may capture the nicked DNA in a partially bent conformation (Figure 5f), analogous to that observed in the "open" conformer of the Lig1-DNA-PCNA complex, where the OBD is flexible (Supplementary Figure 8). The transition from the "open" to the "closed" conformation of the ligase would straighten the DNA and poise it to be encircled by the OBD in the last step of nick sealing.

5. The authors claimed that the interaction between FEN1 and Lig1-OBD contributes to restricting the conformational space of the OBD while this interaction is not direct. This claim should be justified by providing additional evidence. Notably, in the absence of FEN1, the OBD of Lig1 is already located close to the nicked DNA. The authors need to compare between Lig1-PCNA and Lig1-FEN1-PCNA the ratio of the particles showing a stable OBD engaging with the nicked DNA. As FEN1 binds to PCNA through a PIP loop, the authors could consider to generate FEN1 Δ PIP in which the PIP loop is truncated and test if the FEN1 Δ PIP could interact with Lig1 or not.

Following the referee's suggestion, we explored the conformational space of the OBD in the Lig1-DNA-PCNA complex using MD simulations, imposing a restrained distance between the OBD and DNA nick of 5.5 nm (Supplementary Movie 1 and Supplementary Figure 15). Five MD frames corresponding to the most divergent positions of the OBD were extracted and superposed to Lig1 in the Lig1-DNA-PCNA-FEN1 structure (Supplementary Figure 15). The absence of clashes between OBD and FEN1 supports that FEN1 does not restrict the conformational space of the OBD. The discussion has been modified accordingly.

Regarding the ratio of particles mentioned by the referee: the datasets of the Lig1-DNA-PCNA (no ATP) and Lig1-DNA-PCNA-FEN1 complexes did not produce any 3D class

showing a flexible OBD, showing that in the vast majority of the particles the OBD is rigid. We believe that (i) the lack of a constitutive Lig1-FEN1 interface, (ii) the mobility of Lig1 relative to FEN1, (iii) the partial disorder of FEN1 in the toolbelt structure and (iv) the predicted lack of OBD confinement by FEN1, constitute sufficient evidence for the absence of a direct interaction between Lig1 and FEN1 in the toolbelt.

6. If FEN1 could facilitate the OBD binding to the nicked DNA, the ligation activity of Lig1 should be boosted by FEN1. This could be tested by in vitro ligation assay.

As we discussed above, data do not support that FEN1 facilitates OBD binding to the nicked DNA.

Minor points:

1 The archaeal structure (Sverzhinsky et al 2022) should be appropriately introduced and referenced.

Sverzhinsky's paper is now introduced and discussed.

2 Line 127: citations should be given for the neural network algorithm in the main text.

Reference has been added.

3 Line 158: It is not clear how "20% of the flexibility" was obtained. The relevant information was not provided in Supplementary Fig. 6.

The correct figure with relevant information is now indicated in the text (Supplementary Figure 2), thank you.

4 Line 194: The mentioned DNA features should be labeled clearly in the relevant figure.

Figure 1d now shows the relevant labels.

5 Line 504: The detailed strategy for separating the open and close conformations should be provided.

Reconstructions of the Lig1-DNA-PCNA complexes in the open and close conformations derive from the same dataset (Dataset 1, Supplementary Figure 7). A second dataset (Dataset 2, Supplementary Figure 7) was acquired to improve the angular distribution of particles for the *open* reconstruction, and combined with the first dataset. Details of data processing strategy for both conformations are provided in Supplementary Figure 7.

6 In Figure 5, the published structures should be labeled with their corresponding EMDB codes.

PDB code for the Pold-PCNA-DNA-FEN1 toolbelt is now provided in Figure 6 of the current manuscript.

Reviewers' Comments:

Reviewer #1:

Remarks to the Author:

Reading the manuscript and the rebuttal, I am still not convinced that a novel mechanism for how Lig1 is recruited to PCNA is presented.

The authors write in the abstract "We present several cryo-EM structures combined with functional assays, showing that Lig1 recruits PCNA to nicked DNA using two PCNA-interacting motifs (PIPs) located at its disordered N-terminus (PIP_{N-term}) and DNA binding domain (PIP_{DBD}). Once Lig1 and PCNA assemble as two-stack rings encircling DNA, PIP_{N-term} is released from PCNA and only PIP_{DBD} is required for ligation to facilitate the substrate handoff from FEN1."

The authors respond in their rebuttal to reviewer #1 "In the revised manuscript, we provide substantial new experimental data that comprehensively address the mechanism of Lig1 recruitment to PCNA. The new FRET data using an array of Lig variants (Figure 3), combined with the cryo-EM structure (Figure 1), support that Lig1 initially recruits PCNA via a high-affinity PIP motif located at the N-terminus. Once Lig1 and PCNA assemble as two rings encircling DNA, Lig1 is tethered to PCNA via the PIP in the DBD, and the N-terminal PIP is released. "

When reading a review by Timothy Howes and Alan Tomkinson in *Subcell. Biochem* from 2012 (reference 7 in your manuscript), a mechanism for how Lig1 is interacting with PCNA and DNA substrate (recruited to PCNA at a nick) is described based on the available biochemical and structural information 2012. Please explain how their model from 2012 differs from your model? To me, the described mechanism is very, very similar and for that reason I still believe that this manuscript is hampered by a lack of novelty when focusing on the mechanism by which Lig1 is recruited to PCNA.

Furthermore, the presented toolbelt model is attractive but also questioned. Dovrat et al (*PNAS USA* (2014) 111(39):14118-23) found that PCNA heterotrimers containing only one functional binding site enable Okazaki fragment maturation by efficiently coordinating the activities of Pol δ , FEN1, and Lig1. The efficiency of switching between partners on PCNA was not significantly impaired by limiting the number of available binding sites on the PCNA ring. The authors concluded that the "results suggest a mechanism of sequential switching of partners on the eukaryotic PCNA trimer during DNA replication and repair." Thus, the assumed physical handoff between Fen1 and Lig1 may not be required (see model in figure 6).

Reviewer #2:

Remarks to the Author:

Blair et al. adequately addressed most of my questions from the previous round of reviews. However, having now the opportunity to inspect maps and models, I would like to follow up on a few technical comments concerned with the quality of the presented cryo-EM maps and models. Overall, the particle orientation bias documented by a non-uniform coverage of the angular distribution plots and anisotropic 3D-FSCs is evident, to different extent, in all five maps. One of the consequences are distortions in the maps along the preferred orientation axes, undoubtedly contributing to the ambiguity in the molecular models, and inadequate resolution estimates.

For instance, the effects of the directional anisotropy are visible in the Coulomb map of Lig1-DNA-PCNA-ATP complex, captured in the "closed" conformation. Specifically, a part of the map representing PCNA has distortions in the axis perpendicular to the long axis of the map. This fact most likely indicates underrepresentation of the top, bottom, and perhaps tilted views of the complex, with side views predominantly present. Only four class averages are presented in the Supplementary Figure 6a. All four 2D class averages seem to display the side views of the complex. In addition, as previously

acknowledged by the authors, a dip in the FSC curve at around 6 Å may reflect misalignment of particle densities affected by distortions in this particular range of frequencies. Consequently, some structural features in the model do not align with the map, for example some of the beta-sheets in PCNA. Because of the relatively high contour level chosen for the map in the wwPDB EM validation report, it is difficult to comment on its agreement with the model. In my opinion, the manuscript would greatly benefit if the following additional data were included for Lig1-DNA-PCNA-ATP, and for the remaining four reported complexes: (1) Map to model statistics detailing correlation coefficients per residue and per chain for all the map-model pairs. For example, phenix.refine provides such outputs. (2) Representative sets of 2D class averages should be included for each map. Currently, the manuscript only displays four such classes per map. In particular, the top, bottom and side views should be highlighted. Supplementary Fig. 10b shows twelve 2D class averages, however these classes correspond to the map of the "control" deltaN Lig1-DNA-PCNA complex, which is not discussed extensively in the manuscript.

No molecular model was provided for Lig1-DNA-PCNA-ATP complex trapped in the "open" conformation, yet the model is presented in panels b, and d of the Supplementary Figure 8. The validation report for this specific molecular model is also missing. Did the authors deposit atomic coordinates to the RCSB PDB/EMDB? Given the model is shown and described in the manuscript, it should be submitted to the RCSB PDB/EMDB. The map of Lig1-DNA-PCNA-ATP -"open" also suffers from preferred orientations. As the result, it is difficult to interpret the map in the absence of the molecular model, because it is not clear which structural features have been affected by the orientation bias and which have not.

Understandably, due to the low resolution (reported 7.3 Å), no molecular model was derived from the map of Fen1-DNA-PCNA complex. In my opinion, the resolution of this map is severely overestimated. Judging based on the appearance of the structural features in the map, the resolution is likely ~15 Å or less. In addition, the granular pattern in the background of the central map slices, which are presented in the validation report, most likely indicates overfitting. Such overfitting would inflate the resolution estimate.

The map of Lig1-DNA-PCNA-Fen1 looks over-sharpened. What protocols were used for sharpening of the final map? Please include details in the manuscript.

Taken together, distortions of the maps caused by the orientation bias, misalignment of particles, overfitting, and over-sharpening may affect the quality of the structural models, hence influence the interpretation of results. Dealing with the orientation bias is not trivial but authors may consider the following, perhaps in the subsequent manuscripts: (1) increase the tilt angle during data collection. With the state-of-the-art microscope/grid combination set-up employed by authors, the tilt angle could be, in principle, increased from 30 degrees to 45 degrees tilt. (2) Rapid plunging, for example with Chameleon, Vitrojet etc. proved to be useful to limit the orientation bias caused by the adherence of macromolecules to the air-water interface. (3) Finally, the authors may want to investigate a wider range of detergents. CHAPSO was added to samples of Lig1-DNA-PCNA-no ATP complex. Did the authors add CHAPSO to samples representing other four complexes as well? A list of commonly used detergents with the potential for improving the quality of the cryo-EM data by facilitating more complete angular sampling includes, for instance, DMM, DM, Triton X-100, Tween 20 and NP40S.

Reviewer #3:

Remarks to the Author:

I thank the authors for the changes made to their manuscript in response to the comments. Overall, I believe the revised manuscript is suitable for publication.

REVIEWER COMMENTS

Reviewer #1 (Remarks to the Author):

Reading the manuscript and the rebuttal, I am still not convinced that a novel mechanism for how Lig1 is recruited to PCNA is presented.

The authors write in the abstract “We present several cryo-EM structures combined with functional assays, showing that Lig1 recruits PCNA to nicked DNA using two PCNA-interacting motifs (PIPs) located at its disordered N-terminus (PIP_{N-term}) and DNA binding domain (PIP_{DBD}). Once Lig1 and PCNA assemble as two-stack rings encircling DNA, PIP_{N-term} is released from PCNA and only PIP_{DBD} is required for ligation to facilitate the substrate handoff from FEN1.”

The authors respond in their rebuttal to reviewer #1 “In the revised manuscript, we provide substantial new experimental data that comprehensively address the mechanism of Lig1 recruitment to PCNA. The new FRET data using an array of Lig1 variants (Figure 3), combined with the cryo-EM structure (Figure 1), support that Lig1 initially recruits PCNA via a high-affinity PIP motif located at the N-terminus. Once Lig1 and PCNA assemble as two rings encircling DNA, Lig1 is tethered to PCNA via the PIP in the DBD, and the N-terminal PIP is released. “

When reading a review by Timothy Howes and Alan Tomkinson in *Subcell. Biochem* from 2012 (reference 7 in your manuscript), a mechanism for how Lig1 is interacting with PCNA and DNA substrate (recruited to PCNA at a nick) is described based on the available biochemical and structural information 2012. Please explain how their model from 2012 differs from your model? To me, the described mechanism is very, very similar and for that reason I still believe that this manuscript is hampered by a lack of novelty when focusing on the mechanism by which Lig1 is recruited to PCNA.

We thank the reviewer for putting the focus on Howes and Tomkinson’s review. We would like to note that the structural model for Lig1 recruitment proposed in that review is a hypothesis rather than an experimentally validated model, and it also deviates from the mechanism we propose. Howes’ hypothesis stemmed from incomplete biochemical and structural data available at the time, supporting the presence of two PCNA binding sites in Lig1. However, no information on the actual architecture of the Lig1-PCNA-DNA complex existed. While the ligase N-terminal binding site had been characterized in a peptide-PCNA co-crystal structure in the yeast system (Vijayakumar et al, *NAR*, 2007), the DBD binding site had no defined structural or functional characterization. In our work, we provide near-atomic resolution information on the global complex of human Lig1, nicked DNA and PCNA, revealing for the first time the architecture of the functional complex. Our structures allowed us to accurately model the novel PIP-box anchoring the ligase DBD to PCNA. The structures also show that the DBD is the critical

tether of the ligase to PCNA once the two proteins encircle the nicked DNA, and is also critical for DNA handoff from FEN1 for nick sealing. Importantly, differently from Howes' model, we show that the Lig1 N-terminal binding site is an initial tether to PCNA that is released after Lig1, DNA and PCNA assemble as a complex. This has profound functional consequences, as the release of the N-terminal tether makes available one or two PIP-boxes for additional partners' binding to PCNA. In addition, our binding and functional data suggest that the N-terminal tether is redundant in DNA handoff from FEN1. Overall, we believe our data significantly advance the structural/mechanistic understanding of Lig1 recruitment to PCNA for DNA nick localization and most importantly for Okazaki fragment sealing, which is a major focus of our findings. However, we now acknowledge that the flying-cast mechanism for Lig1 recruitment to PCNA has been hypothesized before, with the following statement in the introduction:

“Our results provide further evidence for the previous hypothesis (Timothy Howes and Alan Tomkinson in *Subcell. Biochem*, 2012) of a flying-cast mechanism for Lig1 recruitment, where the high-affinity PIP_{N-term} functions as an initial tether to PCNA. Our structures reveal that, once Lig1 and PCNA assemble as two stack rings encircling the nicked DNA, PIP_{N-term} is released from PCNA and Lig1 stays attached to one PCNA monomer via a low-affinity PIP located in the DBD (PIP_{DBD}).”

Furthermore, the presented toolbelt model is attractive but also questioned. Dovrat et al (*PNAS USA* (2014) 111(39):14118-23) found that PCNA heterotrimers containing only one functional binding site enable Okazaki fragment maturation by efficiently coordinating the activities of Pol δ , FEN1, and Lig1. The efficiency of switching between partners on PCNA was not significantly impaired by limiting the number of available binding sites on the PCNA ring. The authors concluded that the “results suggest a mechanism of sequential switching of partners on the eukaryotic PCNA trimer during DNA replication and repair.” Thus, the assumed physical handoff between Fen1 and Lig1 may not be required (see model in figure 6).

We thank the reviewer for this comment. We have now referenced Dovrat's paper in the discussion, acknowledging the debate on the “toolbelt” versus “sequential” model for the PCNA-directed coordination of Pol δ , FEN1 and Lig1 in Okazaki fragment maturation. We agree that Dovrat's paper has provided biochemical evidence that nick translation, in the yeast system, does not necessarily require simultaneous binding of Pol δ and FEN1 to PCNA, but the rate of nick-translation synthesis in the presence of FEN1 is reduced under sequential model even at long time scales of 0.5 and 1 min (Figure 4B). In addition, the applied methodology has not allowed for evaluation of whether this switching actually occurs. Subsequently, biochemical work characterizing Okazaki fragment processing at the millisecond time scale (Stodola, NSMB, 2016), clearly showed the importance of the formation of a quaternary DNA–PCNA–Pol δ –FEN1 complex to achieve processive nick-translation synthesis. Our own structural work (Lancey, *Nat Commun*, 2020) showed that the human DNA–PCNA–Pol δ –FEN1 complex indeed forms, adding further weight to the toolbelt model in eukaryotic nick translation. In the current work we characterized the structure of the human DNA–PCNA–Lig1–FEN1 toolbelt, and we provided biochemical evidence that Lig1 binding to

PCNA on a preformed DNA–PCNA–FEN1 complex is critical for efficient nick sealing. While our work does not rule out the sequential model in Okazaki fragment processing, it contributes with important novel data supporting the toolbelt model as being more efficient than the sequential model.

Reviewer #2 (Remarks to the Author):

Blair et al. adequately addressed most of my questions from the previous round of reviews. However, having now the opportunity to inspect maps and models, I would like to follow up on a few technical comments concerned with the quality of the presented cryo-EM maps and models. Overall, the particle orientation bias documented by a non-uniform coverage of the angular distribution plots and anisotropic 3D-FSCs is evident, to different extent, in all five maps. One of the consequences are distortions in the maps along the preferred orientation axes, undoubtedly contributing to the ambiguity in the molecular models, and inadequate resolution estimates.

We thank the reviewer for their careful evaluation of our maps and models and their useful comments. Following the referee's suggestions, we have re-evaluated our cryo-EM data. Specifically, we have re-processed the dataset of the open conformer of the Lig1-DNA-PCNA complex obtaining a more isotropic map, and we have modified the model accordingly and deposited it in the Protein Data Bank. The map of the closed conformer of the Lig1-DNA-PCNA complex could not be improved further: considering the limitations of this map and the little novel information of the structure, we have removed the map and model from the manuscript. The maps of the Lig1-DNA-PCNA-FEN1 and FEN1-DNA-PCNA complexes have been sharpened with a different strategy, as detailed further below. Hereafter, we give evidence that the directional anisotropy associated to our maps is not unusual, and we demonstrate, by providing detailed map-to-model correlation statistics, that our models are robust. Thus, we are confident that all biological insight derived from the cryo-EM maps presented in this work is adequately supported by the structural data. All new maps and models are made available to the referee for inspection.

A quantitative evaluation of the anisotropy of the maps presented in the current manuscript version (Supplementary Figures 2d, 6e, 11h and 13d) is provided by the analysis of the directional 2D FSC against spatial frequency using the 3DFSC program (Tan et al, Nature Methods, 2017). Below are presented the 3DFSC plots of the four maps included in the manuscript, together with three exemplars of the same analysis carried out on previously published cryo-EM maps at similar resolutions. From this comparison, it appears clear that the anisotropy present in our maps (in the form of distribution of per angle FSC and standard deviation from the mean of directional FSC) is not unusual. The detailed map-to-model correlation statistics now included in the manuscript further supports that the anisotropy present in our maps does not result in an unsatisfactory correlation between maps and models (see further below).

We calculated the per-chain and per-residues correlation coefficients for all our models (Supplementary Figure 15) using the map-to-model validation tool available in Phenix (Afonine, Acta Crystallogr D Struct Biol., 2018). Below are shown representative plots of the coefficients versus residue number (with the dotted red line representing the average per-chain correlation) for three of our complexes, with a comparison of the coefficients calculated for the recently published 4.4 Å cryo-EM model of *Sulfolobus Sulfataricus* Ligase (ssLig) bound to DNA and PCNA (Sverzhinsky et al, Structure, 2021), which we discuss at length in our paper.

This analysis shows that the overall map-to-model correlation of our models is satisfactory. In particular, the Lig1-DNA-PCNA/noATP model shows excellent

correlation, while the Lig1-DNA-PCNA-FEN1 toolbelt model shows a correlation comparable to that of the archaeal complex for both ligase and PCNA components. The correlation for the FEN1 component (Supplementary Figure 15) is slightly lower, as expected due to the partial disorder of FEN1. The partial disorder/flexibility of FEN1 is acknowledged and explained in the main text (pg. 14). The map-to-model Q-scores in the validation reports are in line with the average Q-scores at the corresponding map resolutions (Pintilie, Nature Methods, 2020; Figure 5).

For instance, the effects of the directional anisotropy are visible in the Coulomb map of Lig1-DNA-PCNA-ATP complex, captured in the “closed” conformation. Specifically, a part of the map representing PCNA has distortions in the axis perpendicular to the long axis of the map. This fact most likely indicates underrepresentation of the top, bottom, and perhaps tilted views of the complex, with side views predominantly present. Only four class averages are presented in the Supplementary Figure 6a. All four 2D class averages seem to display the side views of the complex. In addition, as previously acknowledged by the authors, a dip in the FSC curve at around 6 Å may reflect misalignment of particle densities affected by distortions in this particular range of frequencies. Consequently, some structural features in the model do not align with the map, for example some of the beta-sheets in PCNA. Because of the relatively high contour level chosen for the map in the wwPDB EM validation report, it is difficult to comment on its agreement with the model.

We agree with the referee that the map of the closed conformer of the Lig1-PCNA-DNA complex has some limitations, in particular the poor definition of the ligase Add region including the active site residues, which makes difficult the assignment of this structure to a defined enzymatic step of the nick sealing reaction. Because of these limitations and because the architecture of this complex highly resembles that of the complex obtained without ATP (which as we showed above shows good map isotropy and excellent map-to-model correlation), we have decided to remove the map and model of the closed conformer from the manuscript. The removal of this structure does not affect any of our mechanistic conclusions in Lig1 recruitment to PCNA and modulation of Lig1 function in Okazaki fragment sealing.

In my opinion, the manuscript would greatly benefit if the following additional data were included for Lig1-DNA-PCNA-ATP, and for the remaining four reported complexes: (1) Map to model statistics detailing correlation coefficients per residue and per chain for all the map-model pairs. For example, phenix.refine provides such outputs. (2) Representative sets of 2D class averages should be included for each map. Currently, the manuscript only displays four such classes per map. In particular, the top, bottom and side views should be highlighted. Supplementary Fig. 10b shows twelve 2D class averages, however these classes correspond to the map of the “control” deltaNLig1-DNA-PCNA complex, which is not discussed extensively in the manuscript.

Following the referees' request, per-residue and per-chain correlation coefficients are now presented for all the maps and models presented in the manuscript

(Supplementary Figures 15). In addition, twelve 2D class averages including top, bottom or tilted views are now presented for all presented datasets (Supplementary Figures 2a, 6a, 11d, 13a).

No molecular model was provided for Lig1-DNA-PCNA-ATP complex trapped in the “open” conformation, yet the model is presented in panels b, and d of the Supplementary Figure 8. The validation report for this specific molecular model is also missing. Did the authors deposit atomic coordinates to the RCSB PDB/EMDB? Given the model is shown and described in the manuscript, it should be submitted to the RCSB PDB/EMDB. The map of Lig1-DNA-PCNA-ATP -“open” also suffers from preferred orientations. As the result, it is difficult to interpret the map in the absence of the molecular model, because it is not clear which structural features have been affected by the orientation bias and which have not.

We have reprocessed the dataset of the open conformer of the Lig1-DNA-PCNA complex, obtaining a more isotropic map at 4.2 Å resolution (Supplementary Figures 6 and 7). Because, as we explain in the text, the ligase AdD and DNA are not well resolved in this map, a definitive modelling of the AdD and DNA is not possible. We could however model the PCNA and ligase DBD with good confidence (Supplementary Figure 7). A model of the PCNA-DBD portion of this complex has been deposited in the Protein Data Bank (Supplementary Table 1, Supplementary Figure 15). Modelling of the DBD and PCNA (Supplementary Figure 7) is sufficient to prove that the ligase in this conformer is rotated compared to the complex reconstituted without ATP which shows the DNA nick ends brought together (Figure 1). In the open conformer, the DBD rotation and the encircling of the upstream duplex DNA by PCNA force the DNA nick ends to part. The open conformer structure is important as it shows that nicked DNA can associate to the Lig1-PCNA complex in a partially bent conformation. While the role of the open conformer remains unclear, it may facilitate the handoff of the DNA from FEN1 in Okazaki fragment maturation, or to a repair enzyme in case the nick ends are not ligatable.

Understandably, due to the low resolution (reported 7.3 Å), no molecular model was derived from the map of Fen1-DNA-PCNA complex. In my opinion, the resolution of this map is severely overestimated. Judging based on the appearance of the structural features in the map, the resolution is likely ~15 Å or less. In addition, the granular pattern in the background of the central map slices, which are presented in the validation report, most likely indicates overfitting. Such overfitting would inflate the resolution estimate.

The resolution reported for the map of the Fen1-DNA-PCNA complex in the previous submission was estimated with the gold-standard FSC criterion from the half-maps using the map validation tool in Phenix. However, the map made available to the reviewer in the previous submission was filtered by convoluting with a 3D Gaussian function of width equal to 2σ , for a better visualisation of the partially flexible FEN1-DNA portion of the map (this detail was described in the figure caption of Figure 5 in the previous manuscript). The map included in the current manuscript version was instead

sharpened with anisotropy correction with the `local_aniso_sharpen` tool in Phenix, resulting in a better interpretable map. We now provide an estimation of the map resolution output by Relion (Supplementary Figure 11f), which applies a phase-randomization correction of the FSC (Chen et al. Ultramicroscopy, 2013). The corrected resolution (7.8 Å) is only slightly lower than the one previously reported (7.3 Å). The sharpened map features are consistent with the reported resolution, as the PCNA helices are partially resolved (left panel of picture below, showing a bottom view of the map with fitted model). The FEN1-DNA map portion can be clearly visualized at a lower contour level (right panel of picture below).

Because of the limited map resolution and partial flexibility of the FEN1-DNA subcomplex, we have decided not to deposit a definitive model for this map. However, a detailed procedure on how we generated the model shown in the paper figure (Figure 5b) is presented in the Methods Section, and consists in the simple rigid-body fitting of existing high-resolution structures of the individual components, and prolonging the B-form DNA upstream of the nick through the PCNA pore. This model is now made available to the referee for inspection.

The map of Lig1-DNA-PCNA-Fen1 looks over-sharpened. What protocols were used for sharpening of the final map? Please include details in the manuscript.

We thank the reviewer for pointing this out. The map of the Lig1-DNA-PCNA-FEN1 complex in the previous submission was sharpened using the Density Modification tool in Phenix (Terwilliger et al, Nature Methods, 2020). As the resulting map may appear slightly over-sharpened, we have now performed sharpening directly in Relion, using a global B-factor of -160 \AA^2 .

Taken together, distortions of the maps caused by the orientation bias, misalignment of particles, overfitting, and over-sharpening may affect the quality of the structural models, hence influence the interpretation of results. Dealing with the orientation bias is not trivial but authors may consider the following, perhaps in the subsequent

manuscripts: (1) increase the tilt angle during data collection. With the state-of-the-art microscope/grid combination set-up employed by authors, the tilt angle could be, in principle, increased from 30 degrees to 45 degrees tilt. (2) Rapid plunging, for example with Chameleon, Vitrojet etc. proved to be useful to limit the orientation bias caused by the adherence of macromolecules to the air-water interface. (3) Finally, the authors may want to investigate a wider range of detergents. CHAPSO was added to samples of Lig1-DNA-PCNA-no ATP complex. Did the authors add CHAPSO to samples representing other four complexes as well? A list of commonly used detergents with the potential for improving the quality of the cryo-EM data by facilitating more complete angular sampling includes, for instance, DMM, DM, Triton X-100, Tween 20 and NP40S.

Although above we provided substantial evidence that the anisotropy in our maps does not affect the interpretation of the results, we thank the reviewer for their suggestions on how to further improve the maps. Our tests show that grid tilting beyond 30 degrees is not feasible with our current setup for data collection (AFIS with beam shift/tilt). Modulation of particle orientation by detergents can be effective if holey grids are used, where particles sit in the thin layer of vitreous ice and are susceptible of migrating to the air-water interface. In our case, particles are deposited on a thin layer of graphene oxide embedded in the ice, keeping the particles away from the air-water interface; in this case, detergents do not seem able to affect the particle orientation. Indeed, inclusion of CHAPSO in the buffer did not change the distribution of particle orientation in the case of the Lig1-DNA-PCNA-FEN1 complex.

Description of maps and models made available to Reviewer 2 in this resubmission:

Lig1-DNA-PCNA complex in open conformation:

Model: Lig1_open_conformer.pdb

Maps:

1. Lig1_open_conformer.mrc (consensus reconstruction)
2. Lig1_open_conformer_multibody1.mrc (PCNA body reconstruction after multi-body refinement)
3. Lig1_open_conformer_multibody2.mrc (Lig1-DNA body reconstruction after multi-body refinement)

Lig1-DNA-PCNA-FEN1 toolbelt:

Model: toolbelt.pdb

Map: toolbelt.mrc

FEN1-DNA-PCNA complex:

Model: Fen1_DNA_PCNA.pdb

Map: Fen1_DNA_PCNA.ccp4

Reviewers' Comments:

Reviewer #2:

Remarks to the Author:

The revised manuscript by Blair et al. adequately addresses my concerns raised in the previous round of reviews.

One advice I would give to authors would be to try improving the quality of the models deposited to the RCSB PDB. Further refinement would be of particular importance in the regions involving protein-protein interactions discussed in the paper. This issue could be addressed directly with the RCSB PDB by revising already deposited models.

The Phenix statistics presented for entries with PDB IDs 7QNZ and 7QO1 reveal high clashscore values, significant even for structures at the intermediate resolution. This problem could probably be fixed by a few rounds of the interactive refinements involving Phenix and Coot. Another example is chain A (ligase 1) from the entry with a PDB ID: 8B8T, which displays low atom resolvability and inclusion in the map. This fact is also reflected by low map-to-model correlation coefficients calculated for chain A in Phenix. Again, by employing interactive Phenix and Coot refinements, the authors could possibly improve the validation statistics for the chain A. A similar critique concerns most of the chains from the entry with a PDB ID 7Q01.

Model refinements against the maps with traces of anisotropy at intermediate resolution can pose a challenge, but

one way to improve the models would be by focusing on tracing the backbone in the maps using for example backbone Q-scores as a roadmap, and by removing regions from the models, which in fact have no corresponding densities in the cryo-EM maps, for example some of the loops.

REVIEWERS' COMMENTS

Reviewer #2 (Remarks to the Author):

The revised manuscript by Blair et al. adequately addresses my concerns raised in the previous round of reviews.

One advice I would give to authors would be to try improving the quality of the models deposited to the RCSB PDB. Further refinement would be of particular importance in the regions involving protein-protein interactions discussed in the paper. This issue could be addressed directly with the RCSB PDB by revising already deposited models.

The Phenix statistics presented for entries with PDB IDs 7QNZ and 7QO1 reveal high clashscore values, significant even for structures at the intermediate resolution. This problem could probably be fixed by a few rounds of the interactive refinements involving Phenix and Coot. Another example is chain A (ligase 1) from the entry with a PDB ID: 8B8T, which displays low atom resolvability and inclusion in the map. This fact is also reflected by low map-to-model correlation coefficients calculated for chain A in Phenix. Again, by employing interactive Phenix and Coot refinements, the authors could possibly improve the validation statistics for the chain A. A similar critique concerns most of the chains from the entry with a PDB ID 7QO1.

Model refinements against the maps with traces of anisotropy at intermediate resolution can pose a challenge, but one way to improve the models would be by focusing on tracing the backbone in the maps using for example backbone Q-scores as a roadmap, and by removing regions from the models, which in fact have no corresponding densities in the cryo-EM maps, for example some of the loops.

We thank the Reviewer for their comments. We have re-refined all models included in the paper. The clashscore of all models has been substantially lowered using ISOLDE (Croll, T. I., *Acta Crystallogr. D: Struct. Biol.* 2018). Map-to-model correlation of the PIP_{DBD} binding site of Lig1 in PDB 8B8T has been improved. Refinement and map-to-model correlation statistics has been updated in Supplementary Table 1 and Supplementary Figure 15.